# Flexible TAM requirement of TnpB enables efficient single-nucleotide editing with expanded targeting scope

Xu Feng [1,5] ✉, Ruyi Xu[1,5], Jianglan Liao[1], Jingyu Zhao[1,4], Baochang Zhang[1], Xiaoxiao Xu[1], Pengpeng Zhao[1], Xiaoning Wang[1], Jianyun Yao[2], Pengxia Wang [2], Xiaoxue Wang [2], Wenyuan Han [3] & Qunxin She [1] ✉

TnpBs encoded by the IS200/IS605 family transposon are among the most abundant prokaryotic proteins from which type V CRISPR-Cas nucleases may have evolved. Since bacterial TnpBs can be programmed for RNA-guided dsDNA cleavage in the presence of a transposon-adjacent motif (TAM), these nucleases hold immense promise for genome editing. However, the activity and targeting specificity of TnpB in homology-directed gene editing remain unknown. Here we report that a thermophilic archaeal TnpB enables efficient gene editing in the natural host. Interestingly, the TnpB has different TAM requirements for eliciting cell death and for facilitating gene editing. By systematically characterizing TAM variants, we reveal that the TnpB recognizes a broad range of TAM sequences for gene editing including those that do not elicit apparent cell death. Importantly, TnpB shows a very high targeting specificity on targets flanked by a weak TAM. Taking advantage of this feature, we successfully leverage TnpB for efficient single-nucleotide editing with templated repair. The use of different weak TAM sequences not only facilitates more flexible gene editing with increased cell survival, but also greatly expands targeting scopes, and this strategy is probably applicable to diverse CRISPR-Cas systems.

Programmable RNA-guided nucleases from bacterial and archaeal CRISPR-Cas systems have empowered many fields of biotechnology in the past decade[1–3]. These enzymes can be programmed for specific target recognition and dsDNA cleavage, which provides a basis for their applications in genome editing[4–6]. Efficient target DNA cleavage by Cas nucleases requires a short sequence motif present at either the 3' or 5' flanking the target region known as the protospacer adjacent motif (PAM)[4,5,7]. Structural and biochemical characterization of model Cas nucleases, including Cas9 and Cas12 revealed these enzymes first recognize a cognate PAM sequence before validating the sequence complementarity between the guide and the target sequences[8–10]. These enzymes often show reduced levels of DNA cleavage at targets flanked by non-cognate PAM sequences[11,12], motifs that are normally not considered when choosing targets for genome editing. As a result, the requirement of cognate PAMs has been a constraint on target selections during gene editing[10].

RNA-guided programmable nucleases like CRISPR nucleases have facilitated microbial gene editing in two ways. (1) Target DNA cleavage

[1]CRISPR and Archaea Biology Research Center, State Key Laboratory of Microbial Technology, Shandong University, Qingdao 266237, China. [2]Key Laboratory of Tropical Marine Bio-resources and Ecology, South China Sea Institute of Oceanology, Chinese Academy of Sciences, Guangzhou 510301, China. [3]State Key Laboratory of Agricultural Microbiology, College of Life Science and Technology, Huazhong Agricultural University, Wuhan 430070, China. [4]Present address: College of Life Science, Shandong Normal University, Jinan 250014, China. [5]These authors contributed equally: Xu Feng, Ruyi Xu. ✉e-mail: fengxu@sdu.edu.cn; shequnxin@sdu.edu.cn

in the presence of a PAM in bacterial or archaeal genomes by CRISPR-Cas nucleases, initiates the host's homologous recombination repair with provided repair templates, generating desired mutant cells devoid of targeting[13–15]. (2) These nucleases can also be used in combination with a compatible exogenous recombinase (e.g., lambda Red for *Escherichia coli*) in which the Cas nucleases were used to counter-select against unedited cells[12,16–19]. However, the efficient DNA cleavage with Cas nucleases under both cases may lead to massive cell death and a low transformation efficiency[20–23]. In addition, currently available genome editing tools, including most Cas9 and Cas12 nucleases are tolerant to single mismatches in the target regions[4,7,24–29], therefore these nucleases may efficiently target mismatched sites, hindering their applications in homology-directed single nucleotide editing (SNE). The above barriers have limited the application of CRISPR genome editing in microbial genome engineering[30], encouraging the development of novel programmable RNA-guided genome-editing tools.

Two pioneer studies have reported that bacterial TnpBs encoded by the IS200/IS605 transposon represent a distinct type of RNA-guided DNA endonuclease, from which many, if not all, Cas12 family members have evolved[30–34]. IS200/IS605 transposons are broadly distributed across bacterial and archaeal genomes, and they encode either a transposase TnpA or TnpB of unknown biological functions, or both, depending on different subtypes[35,36]. The transposon is mobilized within a host genome by TnpA that specifically recognizes a conserved short sequence motif for insertion[35,37–39]. Interestingly, characterizations of *Deinococcus radiodurans* TnpB (DraTnpB) revealed that it binds transposon-derived RNA and cleaves dsDNA target flanked by a 5′ transposon/target adjacent motif (TAM) that is the same as the TnpA-recognized sequence[31]. It was thus proposed that TnpA and TnpB may have co-evolved to recognize the same sequence to facilitate the transposon homing process after transposon excision[31,39].

Previously, DraTnpB was shown to induce the formation of small insertions and deletions in mammalian cells[31]. However, whether TnpB can be harnessed for homology-directed genome editing, and the targeting scopes and specificity of TnpB-mediated gene editing remain unclear. In addition, TnpB-encoding genes are among the most abundant genes across bacterial and archaeal genomes[32,33], the activity determinants of TnpB in the natural host have not been explored.

In this work, we investigate activities of TnpBs in *Sulfolobus islandicus* REY15A[40], a model archaeon for studying archaeal biology and biotechnology[41–45]. We find TnpB7 of the IS605-type in this organism recognizes flexible TAM sequences and enables efficient homology-directed gene deletions without inducing apparent cell death. Importantly, the TnpB7 shows a very high DNA targeting specificity on weak TAMs that do not induce cell death and can be harnessed for efficient single-nucleotide editing with greatly expanded targeting scopes in different microbes growing at different temperatures.

## Results

### *Sulfolobus islandicus* IS605-type TnpB enables homology-directed gene editing

To identify TnpB proteins encoded by *S. islandicus* REY15A, the PSI-blast was performed using the DraTnpB protein sequence[31] as the query sequence. A total of 26 protein sequences showing ca. 30% sequence identity to DraTnpB were identified. Eight of them belong to the IS605 group, in which the *tnpB* gene is transcribed in the same direction and partially overlapped with the TnpA-encoding gene[35]. The remaining fall into the IS1341 group that encodes TnpB alone[36] (Supplementary Fig. 1). These two groups of TnpBs are well conserved within each group ( >94% sequence identity) but show considerable sequence divergences in between (sequence identity of ca. 37%). Notably, we found most *tnpB* genes of the IS605 group (namely *tnpB1*

through *tnpB8* according to the gene ID), but not that of the IS1341-type, are associated with a sense-overlapping transcript that extends from the coding sequence region to the 3′ untranslated region of *tnpB* genes (Fig. 1a and Supplementary Fig. 1), which contains a conserved 38/39-nt right end (reRNA) derived sequence and downstream region of variable sequences (Fig. 1b and Supplementary Fig. 2). This result suggests IS605-type TnpBs in this organism may function as programmable RNA-guided nucleases.

Next, we sought to investigate whether the plasmid-borne *tnpB7*-reRNA element could facilitate homology-directed gene deletion using a programmed 24 nt guide targeting the *lacS* gene in the natural host. Two potential TAM sequences were used, including the predicted insertion site of the IS605 transposon in this host, TTTAA (Fig. 1b and Supplementary Fig. 2) and a TTGAT pentanucleotide sequence that supports the RNA-guided DNA cleavage activity of the IS605-type DraTnpB in bacteria[31]. In addition, sequence alignments of IS605 transposon elements in *S. islandicus* REY15A revealed a completely conserved right end ending motif (TTCAC) and a partially conserved motif (TTCACT) (Fig. 1b). For this reason, the 24 nt *lacS* guide sequence was inserted either downstream of the TTCAC or the TTCACT sequence individually, yielding g (0) and g (1) respectively (Fig. 1c). The resulting DNA fragments containing the TnpB7-coding sequence, reRNA-coding region and the guide sequence were cloned into a *Sulfolobus* expression vector, pSeSD[46] or a pSeSD derivative carrying a repair template (RT) containing a mutated *lacS* gene allele, giving two genome-targeting plasmids (g(0) and g(1)) and two gene-editing plasmids (g(0)-RT and g(1)-RT). Resulting plasmids were introduced into hosts in which the *lacS* target region is flanked by either TTTAA or TTGAT, to determine the gene-targeting and gene-editing activities of TnpB (Fig. 1c).

As shown in Fig. 1d, transformation with the g(0) *lacS*-targeting plasmid yielded approximately 4–5 orders of magnitude decrease in colony number in the presence of the TTTAA sequence compared to the non-targeting control plasmid (NT), whereas introduction of the g(1) plasmid into the host yielded no apparent reduction in transformation efficiency. Consistently, the TTTAA motif licences a 79% gene deletion efficiency for g(0), but only a 29% gene deletion efficiency for g(1) (Fig. 1e). These results indicate that TTTAA is a functional TAM for TnpB7 and that the sequence immediately following the UUCAC conserved sequence functions as the guiding sequence. Under these conditions, TnpB7 facilitates efficient homology-directed gene editing in the natural host.

In contrast, the TTGAT motif does not induce any apparent reduction in transformation efficiency regardless of the guide insertion positions (Fig. 1d). Interestingly, it shows an uncompromised gene editing efficiency (74%) with the g(1) guide instead of the g(0) that supports efficient gene editing upon the TTTAA motif (Fig. 1e). Looking into the g(1)-RT construction, we noticed that the "U" following the conserved UUCAC sequence of the reRNA matches the "A" opposite to the last nucleotide of the TTGAT sequence. Given that the guiding sequence starts from the sequence following the UUCAC, we reasoned that the A-U base pairing results in a shifted TAM sequence, TTTGA (Fig. 1e).

Importantly, we found that the gene editing outcome is dependent on the catalytic activity of TnpB7, the TAM sequence and the repair template, and the dropout/mutation of any of these three factors abolished the gene editing activity (Supplementary Fig. 3), suggesting TnpB facilitates TAM- and DNA cleavage-dependent targeted gene deletion via templated DNA repair in the natural host.

### SisTnpB recognizes flexible TAM sequences for target cleavage

The lack of apparent cell death upon the DNA cleavage by TnpB on the target flanked by the TTTGA TAM suggests this sequence represents a weak TAM. To test this idea, we sought to purify the gRNA-TnpB7 protein complex and investigate its DNA cleavage activity on targets

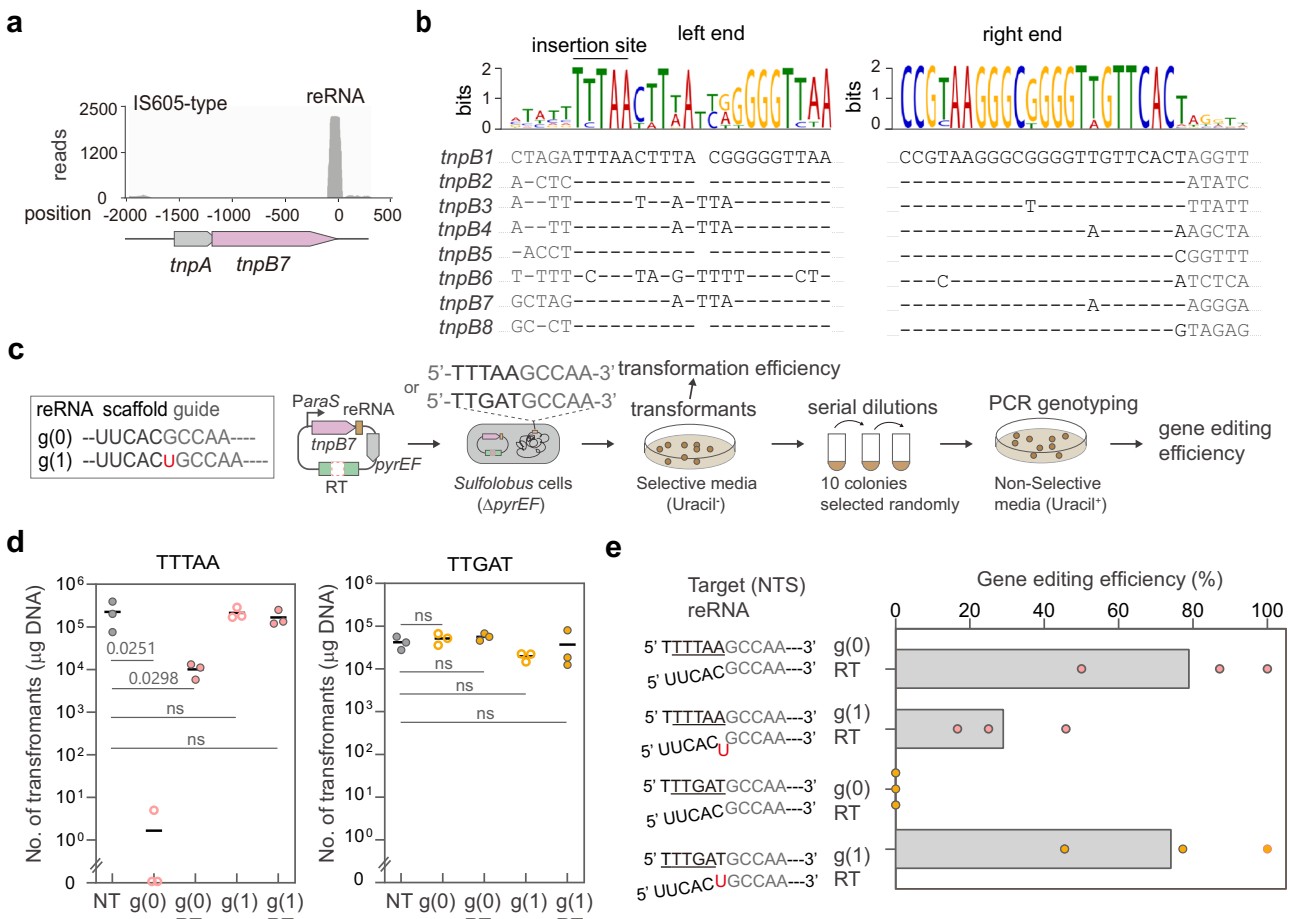

**Fig. 1 | Identification of the guide RNA and TAM required for the gene editing activity of SisTnpB7. a** *sire_2474* gene (*tnpB7*) encoding an IS605-type TnpB is associated with an RNA spanning from the coding sequence region to the right end (reRNA). Reads coverage data were retrieved from the previously published transcriptome data[73]. **b** Sequence alignment of the left end and right end sequences of IS605 transposons of *S. islandicus* REY15A. Conserved sequences are represented by dash lines. The insertion site of the transposon is indicated by the line above the sequence logo. **c** A schematic of different guide RNAs and a workflow of the gene editing experiment. RT refers to the repair template containing the homologous sequences flanking the *lacS* target gene. The reRNAs with the 24 nt *lacS* guide inserted after the most conserved region (UUCAC sequence) and the partially conserved sequence (UUCACU) were defined as the g(0) and g(1), respectively. *pyrEF* is a selection marker for the complementation of uracil auxotrophy. **d** Transformation efficiencies with different TnpB-based plasmids expressing different reRNAs targeting the same target sequence flanked by either the 5' TTTAA or TTGAT sequence. NT is the non-targeting plasmid control. Results of gene targeting/editing plasmids on TTTAA and TTGAT motif are indicated by pink and orange filled circles, respectively. One-way ANOVA was used to compare the means of the NT and other groups based on the data obtained from three independent experiments. p values are displayed above gray lines. The differences between the means are considered statistically significant when the p-value is less than 0.05. "ns" indicates not significant (*p* > 0.05). **e** Gene editing efficiencies of different plasmids upon different TAMs. The left panel shows the inferred base pairing schemes between the guiding sequence and the target region (non-target strand). The inferred TAM sequences are underlined. The bar denotes the data mean of three biologically independent experiments. Results of gene editing plasmids on TTTAA and TTGAT motif are indicated by pink and orange circles, respectively. Source data are provided as a Source Data file.

flanked by either TTTAA or TTTGA TAM. We first purified the TnpB7 protein complexed with the *lacS* guide used for the in vivo gene editing (Fig. 2a). DNA cleavage assays showed that the purified TnpB7 ribonucleoprotein (RNP) efficiently cleaves dsDNA containing the *lacS* target flanked by the TTTAA TAM but does not cleave the unspecific dsDNA substrate bearing no complementarity to the *lacS* guide sequence (Supplementary Fig. 4). Run-off sequencing of the cleaved products revealed a staggered cleavage centered around 15–21 bp from the TAM sequence, yielding 5' overhang (Fig. 2b and Supplementary Fig. 5). In addition, TnpB also showed trans DNA cleavage activities upon the presence of either the ssDNA target or dsDNA target (Supplementary Fig. 4).

Next, we compared TnpB's activities on the dsDNA targets flanked by the two above-discussed TAM sequences. As shown in Fig. 2c, while the TnpB cleaved most of the target DNA in the presence of the TTTAA TAM in 10 min, it took 40 min to cleave a similar amount of target flanked by the TTTGA TAM, indicating the latter is a weak TAM.

To define TAM sequences supporting TnpB's DNA cleavage activity, we systematically examined the activities of TAM variants carrying all possible single nucleotide mutations or different combinations of double mutations and triple mutations. As shown in Fig. 2d, mutation of the TTTAA TAM to TTCAC, the transposon ending motif preceding genomic native TnpB guides (Fig. 1b and Supplementary Fig. 2), abolished the DNA cleavage activity, indicating TnpB does not cleave its native target flanked by the TTCAC motif. In general, mutations of the nucleotide at different positions of the TAM lead to varying degrees of activity reduction (Fig. 2d), indicating the TTTAA sequence is the cognate TAM for TnpB7. However, mutation of the −5T of the TAM to different nucleotides yielded indistinguishable DNA cleavage from the TTTAA TAM, suggesting TnpB7 recognizes NTTAA sequences with similar efficiency. In contrast, nucleotide substitutions at the −3T affected TnpB's activity most. In addition, variation of the −4T generated the second least effect on the activity and mutation of the target-adjacent nucleotide (−1A) yielded the second most effect.

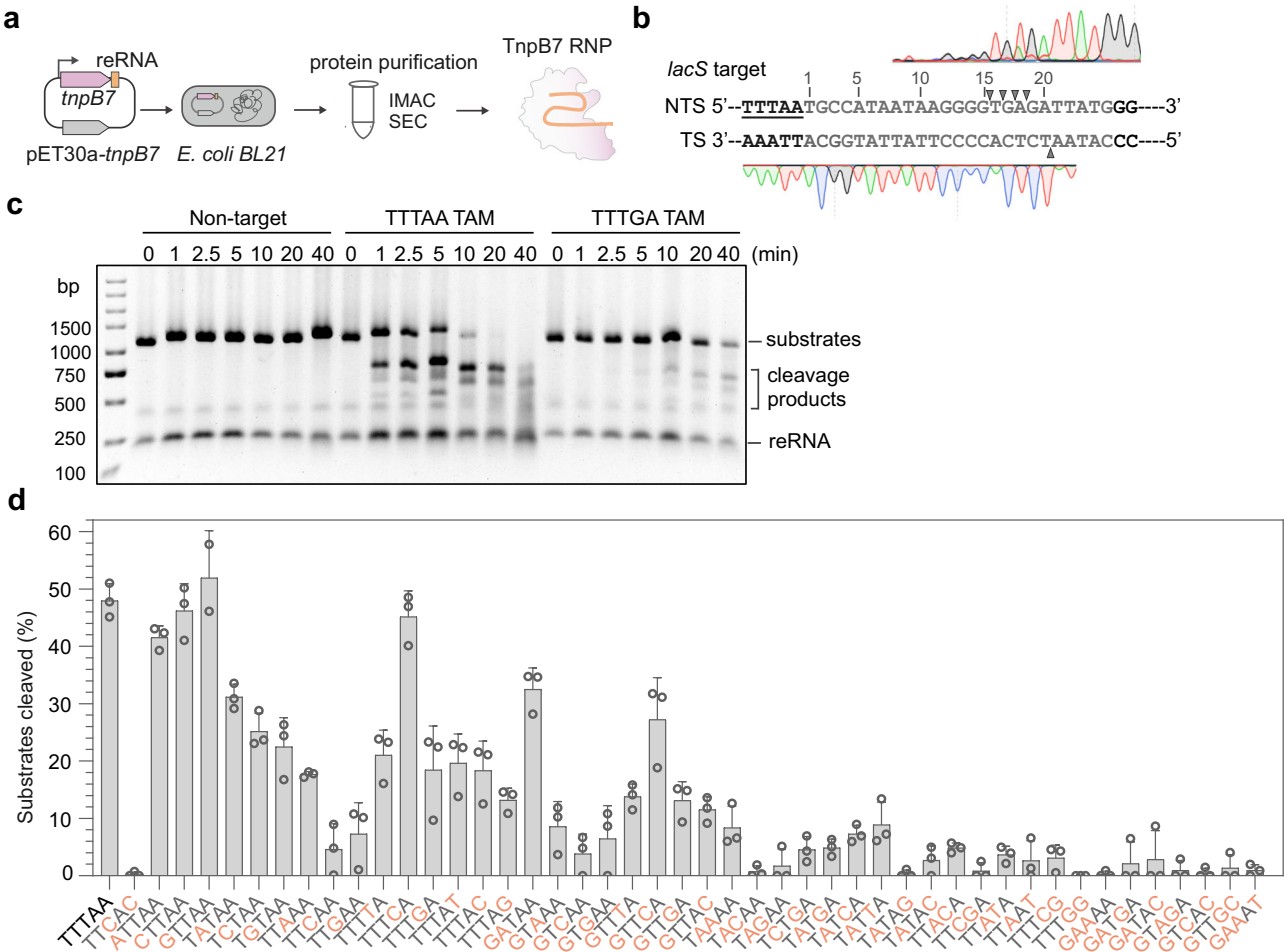

**Fig. 2 | Identification of TAM variants required for the dsDNA cleavage activity of TnpB7. a** A workflow of the co-expression of TnpB7 and the *lacS* guide RNA, and the RNP purification from an *E. coli* host. IMAC, Immobilized metal affinity chromatography. SEC, Size exclusion chromatography. **b** Run-off sequencing result of the cleavage products. Cleavage positions at the non-targeted strand (NTS) and the target strand (TS) are marked by gray triangles. The sequences of the guide-target duplex region are shown in gray and the TAM sequence is underlined. **c** Time-resolved dsDNA cleavage by TnpB on TTTAA and TTTGA TAMs. The reaction system consists of 20 nM dsDNA substrates and 200 nM TnpB RNP complex.

Reactions were incubated at 75 °C for 0, 1, 2.5, 5, 10, 20, 40 min and terminated by mixing with the SDS-DNA loading buffer. Control refers to a substrate that does not contain the *lacS* guide matching sequence. The experiment was repeated three times independently with similar results. **d** dsDNA cleavage activities of TnpB7 on different TAM variant sequences. Each reaction consists of 20 nM dsDNA substrates and 200 nM TnpB RNP and was incubated at 70 °C for 60 min. The value represents the fraction of substrates cleaved by TnpB (defined as the ratio of products to substrates (cleavage products + remaining substrates)). Results are shown as the mean ± SD of three replicates. Source data are provided as a Source Data file.

Interestingly, mutation of the −2A to G or T reduced the activity by approximately 60% compared to that of the cognate TAM, but the mutation of −2A to C barely affected the activity (Fig. 2d). These results suggest that TnpB has a relatively relaxed requirement for the nucleotides at the −5 and −4 positions.

## TnpB facilitates gene editing on diverse TAMs without compromising transformation efficiency

The efficient gene editing activity of TnpB on the target flanked by the TTTGA sequence suggested that other TAM variants may also license efficient gene editing but not induce cell death. To test this idea, we sought to systematically examine the activities of different TAM variants in facilitating targeted gene editing and inducing cell death.

Note that the nucleotide composition of target sites has been shown to affect the targeting efficiency of different RNA-guided nucleases including Cas9 and Cas12[47–50]. In addition, *Geobacillus stearothermophilus* TnpB showed varying levels of DNA interference on different target sequences flanked by the same TAM[39]. Therefore, target site selection at different genomic locations may also affect the activity of SisTnpB, thus interfering with the activity comparison

between different TAMs. To exclude effects of the nucleotide composition of different target sites on the activity of SisTnpB, we constructed a series of mutant strains in which the *lacS* target flanking sequence was mutated to the in vitro characterized TAM sequences corresponding to different levels of DNA cleavage (Figs. 3a and 2d). Then these strains were used to test the DNA interference activity of TnpB in vivo, i.e. the reduction in the transformation efficiency with the *lacS*-targeting plasmid (g*lacS*), and the gene editing efficiency with the g*lacS*-RT plasmid.

As shown in Fig. 3b, the introduction of the *lacS*-targeting plasmid into S1, S2 and S3 strains individually decreased transformation efficiency by more than 1000 folds, demonstrating that TnpB targeting at targets flanked by those TAM sequences corresponding to an in vitro DNA cleavage activity of 27–48% induced massive cell death. Notably, transforming S1, S2, and S3 with the gene-editing plasmid (g*lacS*-RT) yielded a 100‑1000-fold higher transformation efficiency compared to that with the gene-targeting plasmid (g*lacS*), indicating plasmid-borne repair templates may have facilitated the repair of TnpB-introduced DNA breaks. Consistently, we found that TnpB7 facilitated efficient *lacS* gene deletion in these strains (Fig. 3a), reinforcing that

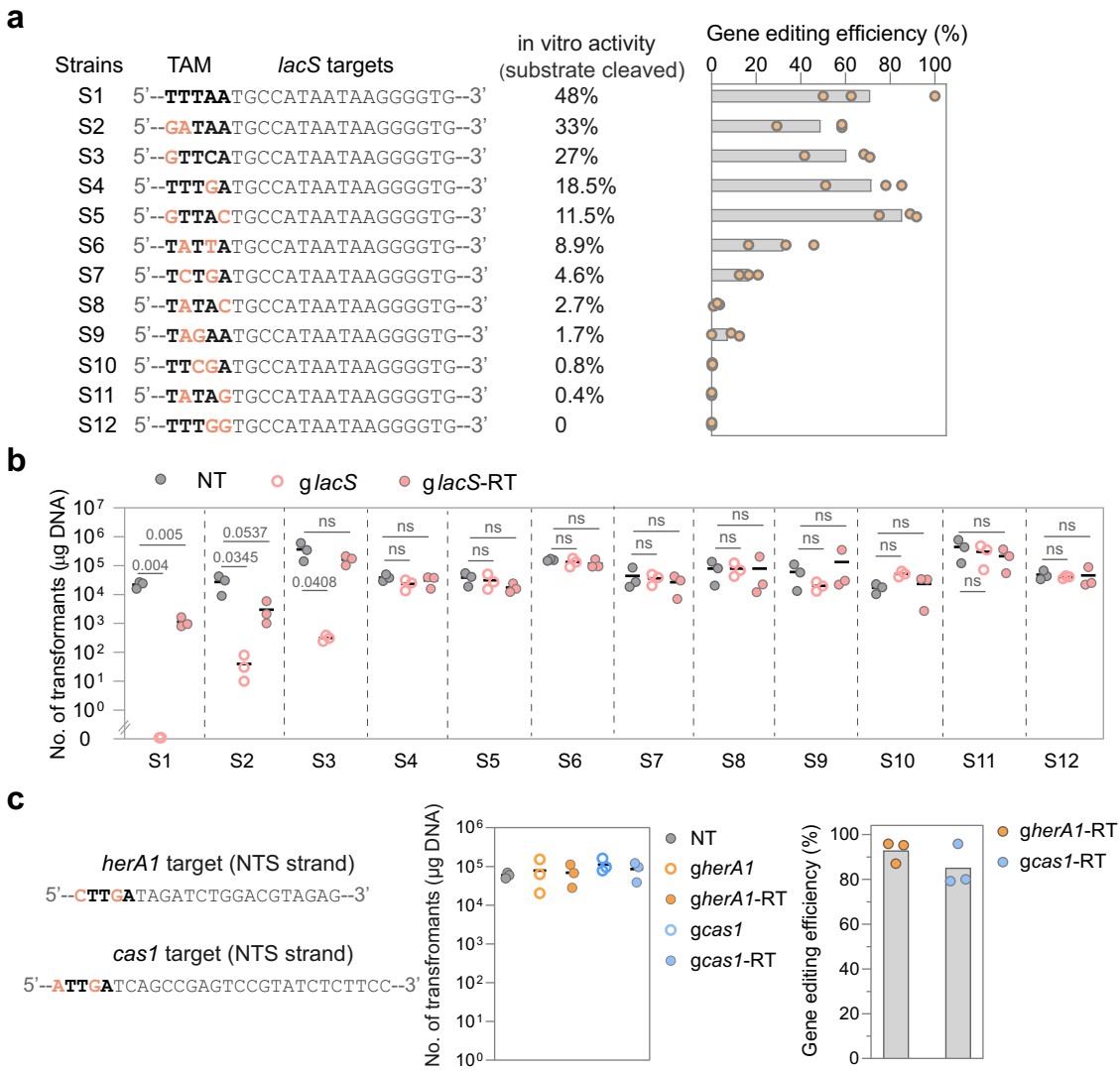

**Fig. 3 | TnpB7 facilitates gene editing on targets flanked by flexible TAM sequences. a** The left panel shows a diagram of target sites in different strains. Note that only the sequences of the non-target strand are shown. Mutated nucleotides at the TAM region are highlighted in orange. TnpB's DNA cleavage activity on each TAM variant is indicated (as determined in Fig. 2d). The right panel indicates gene editing efficiencies of the *lacS*-editing plasmid (g*lacS*-RT) in different strains. The bar denotes the data mean obtained from three biologically independent experiments. **b** Transformation efficiencies with TnpB-based plasmids with different strains. NT means the non-targeting plasmid. g*lacS* means the *lacS*-targeting plasmid, and g*lacS*-RT refers to the *lacS* gene-editing plasmid. The transformation efficiency was defined as the colony formation unit of electroporated cells per 1 μg

plasmid DNA. One-way ANOVA and then Tukey test were used to compare the means of the NT and other groups based on the data obtained from three independent experiments. p values are displayed above gray lines. The differences between the means are considered statistically significant when the p-value is less than 0.05. "ns" indicates not significant ($p > 0.05$). **c** The gene editing activity of TnpB on other genomic targets flanked by weak TAM sequences. The left graph shows the transformation efficiencies with gene-targeting plasmids and gene-editing plasmids. NT means the non-targeting control. The right graph shows the gene-editing efficiency results. Data were obtained from three biologically independent experiments. Source data are provided as a Source Data file.

target sites flanked by strong TAM sequences can be used for gene editing.

In contrast, transforming S4 strain with the *lacS*-targeting plasmid yielded indistinguishable cell counts from that with the NT plasmid (Fig. 3b), suggesting DNA cleavages generated at target sites flanked by those TAMs supporting a DNA cleavage below 20% can be repaired in time without interfering with cell growth. Importantly, TnpB facilitated the *lacS* gene editing at the target flanked by the TTTGA TAM with a similar efficiency as that flanked by strong TAMs (Fig. 3a). In addition, GTTAC TAM with an 11.5% in vitro activity also licenced a gene editing efficiency of >80%, confirming that the sequences that are associated with an activity range of 11.5–18.5% are competent for gene editing but not inducing apparent cell death (Supplementary Table 1).

In addition, we found that TATTA, TCTGA and TAGAA TAMs associated with 1.7–8.9% in vitro activity also facilitated gene editing, but with reduced efficiencies. In contrast, TTCGA, TATAG and TTTGG that are with < 1% DNA cleavage activity failed to facilitate gene editing (Fig. 3a), which is very likely due to that residual activities of TnpB on these sites were not sufficient to confer gene editing to a detectable level. To facilitate gene editing at target sites flanked by the weak TAM with limited activities, we performed the gene editing experiment in the presence of arabinose that increases the expression strength of the P*ara*S promoter (by 6-fold) driving the expression of the TnpB and guide RNA. As shown in Supplementary Fig. 6, while TnpB only facilitated less than 10% gene editing on the TCTGA TAM under the basal expression level, it drove gene editing with an efficiency of more than

40% under the induced condition, indicating weak TAMs can also be used for the gene editing by tinkering with the expression of effectors.

To test whether these TAMs could mediate the gene editing at other genomic locations, two target sites at the *herA1* and *cas1* genes flanked by CTTGA and ATTGA sequences, respectively that are associated with an approximately 14% DNA cleavage activity (Fig. 2d) were chosen. As shown in Fig. 3c, while transformation with these gene-targeting and gene-editing plasmids did not decrease transformation efficiency, targeted gene deletions on the *herA1* and *cas1* targets with an efficiency of 91% and 82%, respectively were observed. These results further confirmed that TnpB7 can drive efficient homology-directed gene deletion without compromising transformation efficiency when targets are flanked by weak TAMs.

### Revealing the minimal guide RNA requirement of TnpB7-based gene editing

Comparative structural analyzes between DraTnpB and Cas12 revealed that the RNA components of TnpB may have substituted the function of certain protein domains of Cas12[34,51]. Consistently, TnpB proteins are associated with relatively large RNA components compared to Cas12 members[34,51–53]. We then sought to determine the minimal RNA component essential for the gene editing activity in the TnpB gene-editing system. Given the reRNA-coding sequence and the TnpB-coding sequence are overlapped, truncation analysis of the reRNA element would also truncate the *tnpB* gene. To avoid such a scenario, a dual-plasmid system recently developed for this archaeon[54] was employed in which, one plasmid was used to express TnpB7 protein, and the other, for producing a guide RNA (reRNA) of variable sizes and for providing the repair template. As shown in Supplementary Fig. 7b, co-transformation with the TnpB7-expressing plasmid (pTnpB) and the 166 or 116 (differs in the length of TnpB-coding sequence region) guide RNA-expressing plasmid yielded 10–20% *lacS* edited colonies (white colonies), indicating TnpB can function together with a guide RNA transcribed at a different location. Interestingly, the gene editing efficiency of the dual-plasmid system is lower than that of the single gene editing plasmid targeting the same target. However, the decrease in editing efficiency is diminished once the *tnpB* and the reRNA were expressed from the same plasmid, even when they were expressed as distinct transcripts (Supplementary Fig. 7c), indicating the way how reRNA was expressed affected the in vivo activity of the TnpB effector complex. In addition, the introduction of an archaeal transcriptional terminator[55] immediately downstream of the guide sequence significantly increased the gene editing activity (40% vs 16% and 34% vs 10%) (Supplementary Fig. 7b), suggesting the 3' padding sequence of the reRNA may have interfered with the function of the reRNA probably by forming intramolecular secondary structures with the guiding sequence. Using this dual-plasmid gene editing system, we also revealed that truncation of the gRNA from the 5' terminal to less than 113 nt (CDS-derived region) leads to dramatically reduced activities (Supplementary Fig. 7b), demonstrating that the full length of reRNA required for TnpB gene editing is approximately 176 nt.

Next, we sought to determine the minimal length of the guide-target matching region required for the gene editing activity of TnpB. To this end, the 25 nt *lacS* guide sequence of the TnpB gene editing plasmid was truncated from the 3' terminal to indicated lengths (Fig. 4a), and the gene editing efficiency of each plasmid carrying different guide sequences was assayed with different TAMs. As shown in Fig. 4a and Supplementary Fig. 8, upon non-cognate TAMs (TTTGA and GTTCA), a guide longer than 14 nt supported a similar activity of TnpB as the 25 nt guide sequence. The 14 nt guide sequence slightly affected the gene editing activity, and shorter guide sequences (less than 14 nt) eliminated the activity. Differently, we found that a 13 nt guiding sequence licenced gene editing upon the cognate TAM (TTTAA) (Fig. 4a). These results indicate the minimal requirement of

guide RNA-target duplex region required for the gene editing activity with TnpB7 is 13–14 bp.

Cas9 and Cas12 nucleases are generally tolerant to mismatches in the target region, which has hindered the application of these tools in single-nucleotide genome editing[25,56–58]. To test the DNA targeting specificity of the TnpB system, we prepared 15 guide RNA variants by tiling 1 nt mismatches across the 15 nt *lacS* guide region and analyzed the activity of resulting mutated guide sequences in driving targeted gene deletion on the targets flanked by either the TTTGA or TTTAA TAM. As shown in Fig. 4b, any mismatch in the 15 bp guide-target matching region dramatically reduced the gene editing efficiency, indicating successful gene editing on weak TAMs by TnpB requires full sequence matching between the guiding sequence and the DNA target. Interestingly, when the target was flanked by the TTTAA sequence, robust gene editing and reductions in transformation efficiency were observed with mismatched gRNA (Fig. 4b and Supplementary Fig. 9), meaning that TnpB can facilitate gene editing at mismatched targets in the presence of the cognate TAM. To investigate how the mismatches in the target region affect TnpB's activity, we compared the in vitro DNA cleavage activities of TnpB on matched targets and mismatched targets flanked by either TTTAA or TTTGA TAMs. As shown in Supplementary Fig. 10, TnpB cleaves the dsDNA targets containing mismatches at the 1st, 5th and 9th position with reduced efficiency compared to the full-matching target flanked by TTTGA TAM, indicating TnpB is sensitive to mismatches at the duplex region upon weak TAMs. Interestingly, TnpB cleaves the substrate containing a mismatch at 5th bp from the TTTAA TAM with uncompromised efficiency as the full-matching target. These results suggest the DNA cleavage activity of TnpB on mismatched targets flanked by the TTTAA but not the TTTGA is still sufficient for gene editing.

### TnpB enables flexible homology-directed single-nucleotide editing on weak TAM sites

The high targeting specificity of TnpB on the weak TAM suggests this system can be harnessed for homology-directed single-nucleotide editing (SNE). To test this idea, we constructed SNE plasmids targeting the *lacS* gene, of which the corresponding repair templates contain a single nucleotide mutation across the guide-target matching region, and analyzed the transformation efficiency and/or gene editing efficiency of TnpB with these SNE plasmids on the target flanked by the TTTAA or TTTGA TAMs. The results show that transformation with the SNE plasmids yielded a similar colony number compared to that of the non-targeting plasmid when the target is flanked by the TTTGA (Supplementary Fig. 11). Importantly, DNA sequencing results of colony PCR products revealed that ca. half of the colonies or more contain mutations at different positions of the guide-target duplex region as the RT (Fig. 5a).

We then tested the weak TAM-facilitated SNEs on other genes. Targets flanked by ATTGA and CTTGA weak TAMs were chosen from the *cas1* and *herA1* genes, respectively and subjected to SNE (Supplementary Fig. 12). Again, the results revealed that transformation with gene-editing plasmids targeting the *herA1 or cas1* generates approximately 80 - 90% of edited cells without inducing apparent reductions in transformation efficiency, which is comparable to the efficiency of TnpB7 in mediating gene deletion at the same sites (Fig. 3c). These results reinforced that TnpB enables efficient and flexible SNE at targets flanked by weak TAMs.

In contrast, when the target is flanked by TTTAA, the transformation efficiency with the SNE plasmids was >10000 times lower than that with the non-targeting plasmid (Supplementary Fig. 13). In fact, no colonies were obtained in two of the three replicates experiments. The possible reason could be that the strong TAM facilitated efficient cleavage of the wt genome, the edited genome and even the repair template by the TnpB, such that very few cells survived on the selective medium.

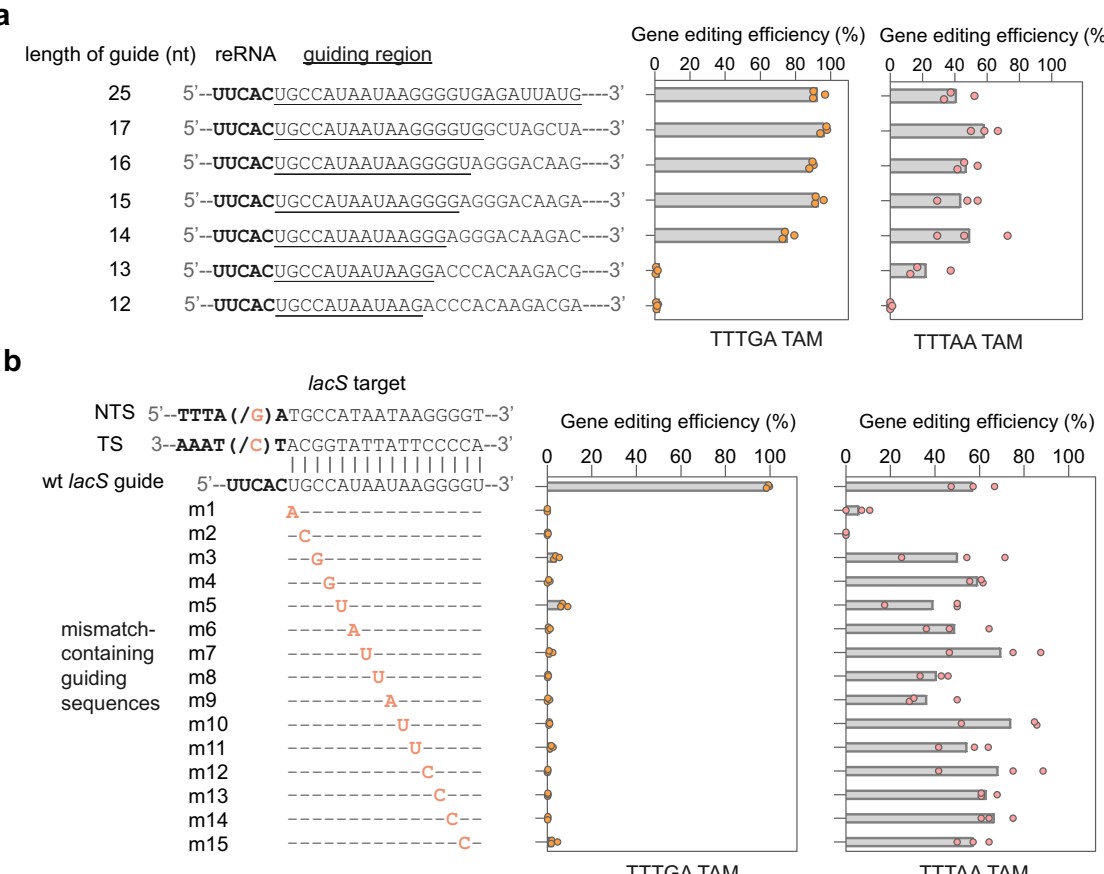

**Fig. 4 | Minimum requirements of the guiding sequence required for gene editing with TnpB. a** The minimal length of the *lacS* guiding sequence that supports gene editing activity of TnpB on the targets flanked by different TAMs. The *lacS* guide sequence was truncated from the 3′ terminal to the indicated lengths. The guide-target duplex regions are underlined. Data are obtained from three independent experiments and the bar denotes the data mean. **b** The effect of single base transversion in a 16 nt *lacS* guiding sequence on the gene editing efficiency of TnpB7. Mutated nucleotides in the TAM sequence or the guiding region are highlighted in orange. The experiment was performed on the target site flanked by either the TTTGA or TTTAA TAM. The bar denotes the data mean of three biologically independent experiments. Source data are provided as a Source Data file.

Next, we asked whether strong TAMs can be used for SNE editing by reducing the expression level of the effector complex. To this end, we replaced the original P*araS* promoter driving the expression of the TnpB and reRNA with P*araS*-m38[59], a mutated version with ca. 1/8 strength of the original one (Supplementary Fig. 14), and analyzed the gene editing outcomes of the mutated SNE plasmid in the presence of the TTTAA TAM. Upon the reduced expression of TnpB, transforming cells with the SNE plasmid showed >1000-fold higher transformation efficiency, indicative of much-reduced DNA targeting activities (Fig. 5b). However, reduced expression of TnpB still facilitated equally efficient gene deletion as the TnpB driven by the original promoter, suggesting the activity of TnpB is more than sufficient for efficient gene editing when expressed using the P*araS* promoter. Importantly, TnpB also enabled SNE editing on more than half of the colonies without inducing extensive cell death (Fig. 5b), indicating that TnpB also facilitates SNE upon strong TAMs after tinkering with the expression of the effector.

While SNE editing can be achieved either by reducing the expression of TnpB with the strong TAM or simply using weak TAMs, the question remains as to whether the target flanked by the weak TAM is more sensitive to single mismatches than the target with the strong TAM if the activities of TnpB on the two sites were similar. To test this, we sought to enhance the DNA cleavage on the weak TAM by elevating the TnpB expression. Then we compared the DNA targeting activities of TnpB on different TAMs. As shown in Fig. 5c and Supplementary Fig. 14, increasing the expression of TnpB (by about 6-fold) elicited an apparent DNA interference on the weak TAM site to a similar (or stronger) level as that seen on the strong TAM (under the non-induced level). Interestingly, the introduction of a repair template containing the single mismatch increased the colony numbers from 0 to more than 3700 in the case of the weak TAM, suggesting the mismatched repair template rescued the cells from consistent targeting by providing a repair template devoid of targeting. In contrast, The introduction of the same mismatched repair template only increased colony from 3.3 to 40 for the target with the strong TAM, suggestive of consistent targeting on the mismatched target. These results indicate that upon the strong TAM, TnpB shows more tolerance to mismatches at the guide-target matching region than upon weak TAM sites.

### TnpB7 facilitates efficient genome editing in different bacteria
Given the efficient gene editing activity of TnpB in the natural host, we asked whether this thermophilic archaeon-sourced protein could function in unrelated bacterial hosts that grow at relatively low temperatures. We firstly tested the activity of TnpB7 in the model bacterium, *E. coli*, growing at 37 °C, as TnpB7 was shown to form functional RNP in this host (Fig. 2). To facilitate the quantification of gene editing efficiencies, we aimed to mutate part of the *lacZ* gene into a fragment containing a stop codon and an EcoRI site. Successful gene editing can be visualized by X-gal staining of colonies and subsequently verified by enzyme restriction digestion of the PCR amplicon (Fig. 6a). A total of four target sites flanked by either strong or weak TAM sequences were selected (Fig. 6b). Interestingly, neither the

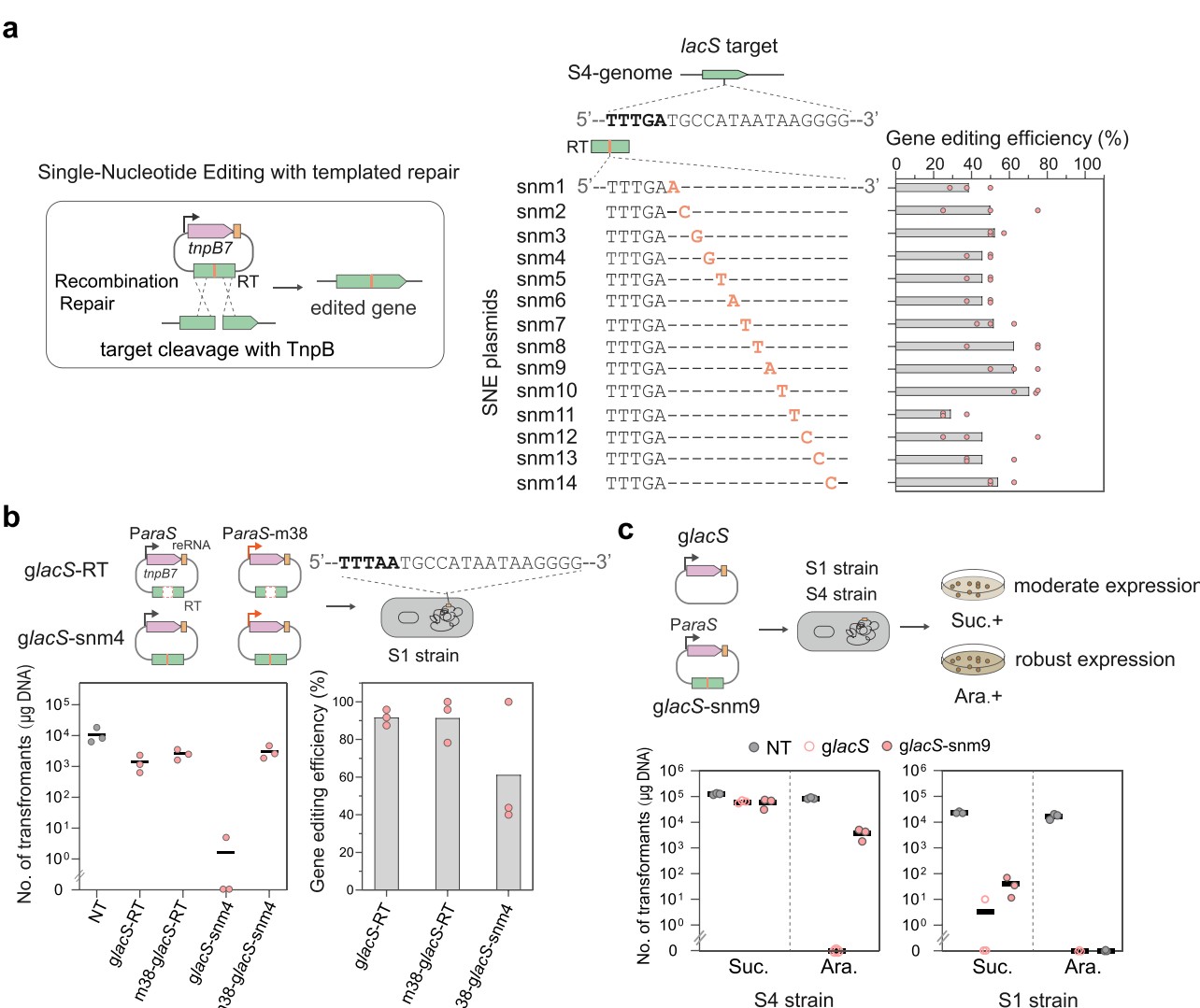

**Fig. 5 | TnpB facilitates flexible single-nucleotide genome editing with templated repair. a** Gene editing efficiencies of TnpB-based SNE plasmids on the *lacS* target flanked by the TTTGA TAM. Mismatched positions between the *lacS* target and repair templates are indicated by nucleotide sequences in orange. Only the sequences of the NTS of the target region are shown. The bar denotes the data mean of three biologically independent experiments. **b** Gene editing outcomes with TnpB upon the reduced expression at the target flanked by the TTTAA TAM. P*araS*-m38 is a P*araS* promoter derivative with a reduced expression level. NT means non-targeting control. The column denotes the data mean. **c** Comparative analysis of SNE outcomes of TnpB on weak and strong TAM sites. Suc. and Ara. refer to sucrose and D-arabinose, respectively. Data were obtained from three biologically independent experiments. Source data are provided as a Source Data file.

gene-editing plasmids (g*lacZ*-RT) nor the genome-targeting plasmids (g*lacZ*) caused a reduction in transformation efficiency on all four target sites compared to the vector control, probably due to a reduced cleavage activity of the thermophilic protein at the unfavorable temperature (37 °C). However, successful gene editing (35–60% efficiency) was detected for the gene editing plasmids targeting all four targets including those flanked by weak TAMs. Furthermore, TnpB7 also facilitates homology-directed single-nucleotide editing on the *lacZ2* and *lacZ3* targets flanked by TTTAA and TTTGA TAM, respectively, with an efficiency of ca. 50% (Fig. 6b), confirming that TnpB also enables flexible gene editing in unrelated bacterial hosts.

Next, we tested whether the TnpB could be used for gene editing in *Vibrio alginolyticus*, one of the most important pathogenic bacteria of worldwide distribution infecting marine animals[60]. As illustrated in Fig. 6c, the gene editing plasmid was delivered to *V. alginolyticus* by conjugation assisted by the *E. coli* WM3064 strain. Among the randomly selected colonies from three replicate experiments, approximately 80% of colonies were proven to be gene-edited at the target site

of the *V. alginolyticus dgc137* gene, as revealed by the PCR genotyping results (Fig. 6c and Supplementary Fig. 15). As the strain was grown at 30 °C, this confirmed the TnpB7 can be used as a genome editor at temperatures ranging from 30 °C to 75 °C, thus holding the promise of being developed as a universal gene editing system for both thermophilic and mesophilic microbes.

## Discussion

In this study, we demonstrate that an archaeal RNA-guided programmable nuclease TnpB recognizes a broad range of TAM variants beyond the cognate TAM for efficient homology-directed genome editing without inducing apparent cell death. The use of weak TAM sequences not only greatly expands the targeting scope, and increases the numbers of gene-edited colonies, but also enables flexible single-nucleotide editing with templated repair (Fig. 7). Our finding might be extended to diverse RNA-guided systems, including different TnpB nucleases[52,61,62], Cas12 nucleases and recently identified eukaryotic Fanzor nucleases that are closely related to TnpB[31–33,63].

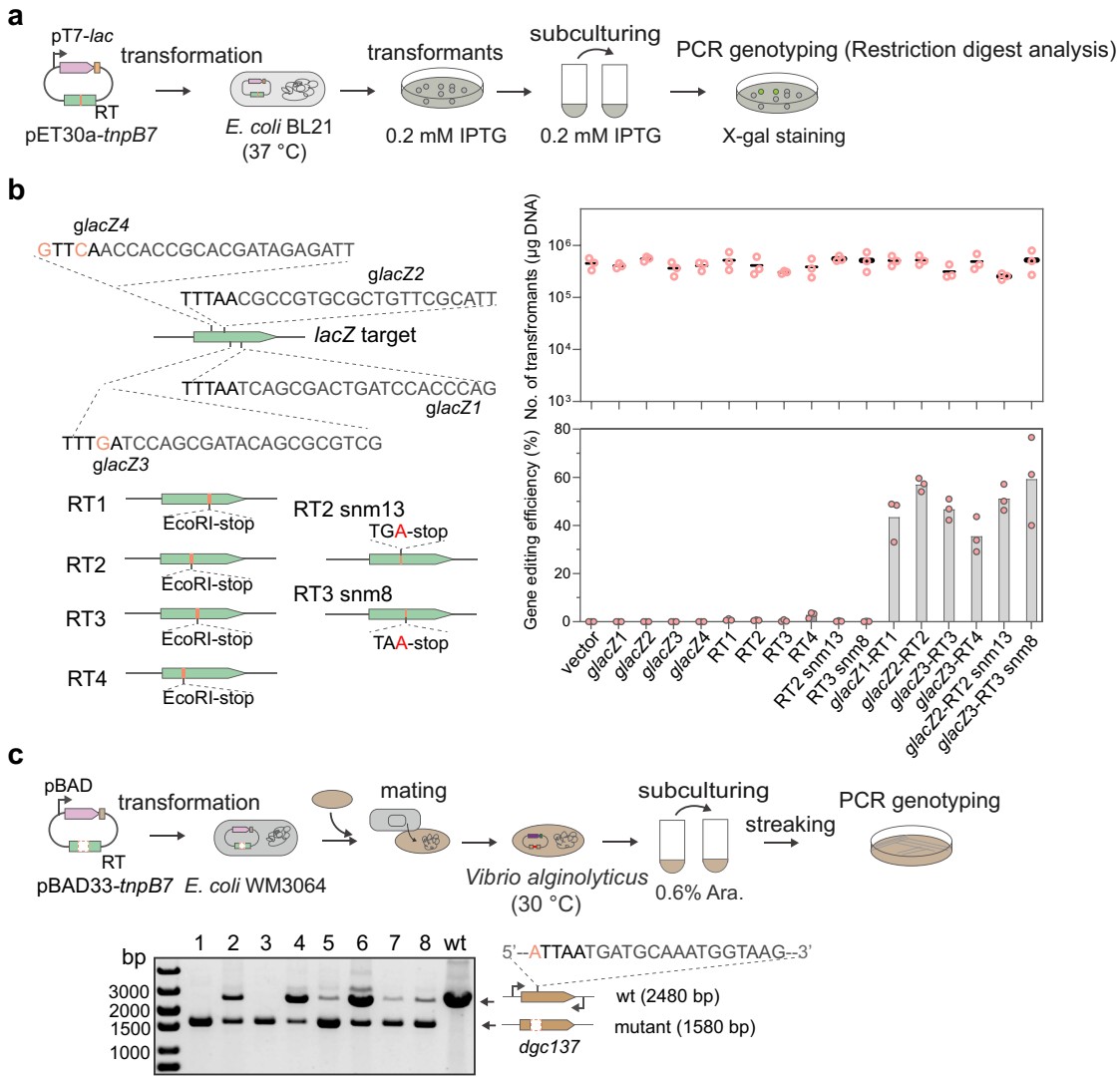

**Fig. 6 | TnpB enables flexible genome editing in different bacteria. a** A workflow of gene editing assay with TnpB7 in *E. coli*. RT means a repair template. The genotypes of colonies were determined by the X-gal staining and the colony PCR. **b** Gene editing outcomes with TnpB in *E. coli*. RT1 to RT4 indicate the repair templates containing the mutated target sites. RT2-snm13 and RT3-snm8 refer to the repair template containing a single-nucleotide mutation at the 13th and 8th bp from the corresponding TAM sequence, respectively. Columns indicate the means of the editing efficiencies determined from three independent experiments. Source data are provided as a Source Data file. **c** Gene editing assay with TnpB7 in *V. alginolyticus*. One of representative images from three biological independent experiments is shown. Lane 1 to 8 of the gel image represents the colony PCR results for 8 randomly selected colonies from the streak plate. wt, wild type control.

The identification of diverse weak TAM variant sequences in supporting efficient gene editing suggests TnpB's DNA cleavage activity on the cognate TAM (TTTAA) site is more than enough for driving gene editing in host cells. Upon targeted DNA cleavage with TnpB, since no non-homologous end-joining DNA repair pathway has been reported in *S. islandicus*[41], the host cells presumably employ the homology-directed repair with plasmid-borne repair templates or the sister chromosome to repair the TnpB-introduced DNA breaks. It is likely the DNA damage resulting from the reduced DNA targeting on targets flanked by weak TAMs can be consistently repaired in time by the DNA repair machinery without interfering with cell growth. Therefore, efficient gene editing with enhanced cell survival can be achieved either by reducing excess DNA cleavage activities in the cells or by elevating the host DNA repair capacity (e.g. introduction of a compatible exogenous recombination repair system or elevating the expression of host DNA repair proteins). Consistent with our results, a recent study has revealed that attenuating the DNA targeting activity of Cas9 or Cas12a in bacteria boosts the transformation efficiency without compromising gene editing efficiency under certain cases[64].

Since most TAM/PAM variant sequences supporting a reduced DNA targeting activity that is outcompeted by the host DNA repair capacity do not induce apparent DNA interference, these sequences can be overlooked by the in vivo plasmid depletion assay, a frequently used strategy to identify the PAM sequences for diverse Cas nucleases, based on their ability in depleting target plasmids upon strong PAMs[7,12,62,65–68]. Therefore, our results hint at the possibility that the PAM/TAM sequences of different RNA-guided systems required for in vitro DNA cleavage, in vivo plasmid clearance (or chromosome-targeting), and gene editing are different, partly overlapped (Fig. 7), and vary between different hosts in which the expression level of the effector complex and DNA repair capacity are host-specific. All these factors should be considered for on-target and off-target gene editing predictions.

Flexible SNE with TnpB on targets flanked by weak TAM and by strong TAM under reduced expression level (Fig. 7), on the one hand, may partially be attributed to the reduced DNA targeting activities. Indeed, it has been shown that reducing the active Cas9 amount or shortening the guide RNA length in mammalian cells facilitates the

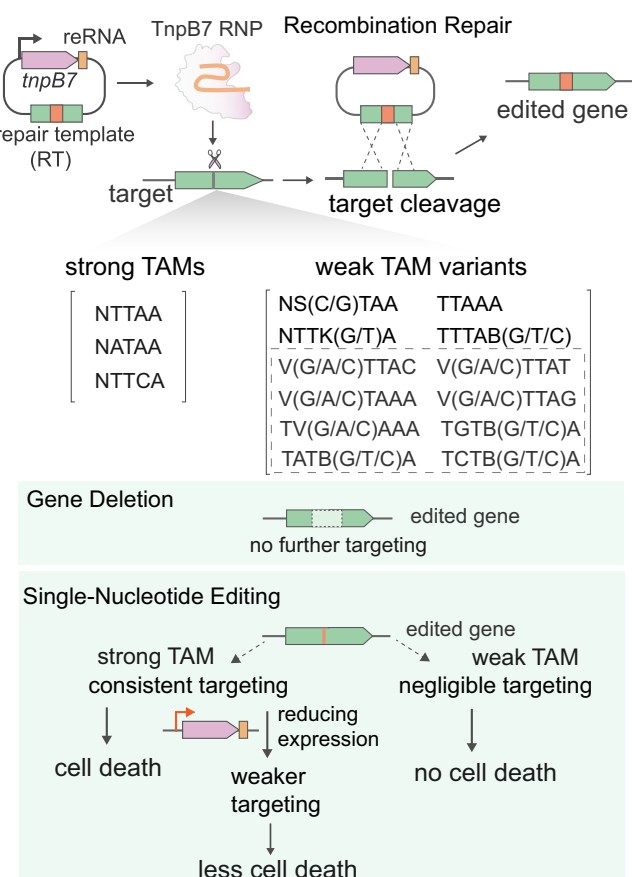

**Fig. 7 | TAM-flexible gene editing with TnpB.** The figure depicts the relationship between targeted DNA cleavage activities of TnpB with different TAM variants and gene editing outcomes. The flexible TAM requirement of TnpB enables efficient gene deletion on strong and weak TAM sites, which greatly expands the targeting scope. However, in the case of strong TAM sites, TnpB efficiently cleaves targets containing single-nucleotide mismatches, leading to cell death. In contrast, the cleavage on the target sites or mismatched targets flanked by weak TAMs can be efficiently repaired, therefore it does not yield apparent cell death. This feature can be leveraged for efficient single-nucleotide editing with templated repair. Nevertheless, by tinkering with cellular activities of TnpB to a level that is not overwhelming to the host repair capacity, TnpB can also achieve single-nucleotide editing on strong TAM sites without inducing massive cell death. RT refers to repair template devoid of a target site or containing a mismatched target site. TAM sequences associated with an in vitro cleavage activity higher than 27% were defined as strong TAMs. TAM variants of limited gene deletion efficiencies (with an activity lower than 11.5%) are highlighted by light gray letters, the efficiencies of these TAMs in mediating gene editing can be enhanced by elevating the expression level of TnpB.

targeting specificity[27,69–71] and Cas9 can be used to introduce single nucleotide mutations in bacteria with target-mismatched sgRNAs[25]. Since these strategies normally lead to reduced DNA targeting, this suggests reducing the cellular targeting activities could be a shared strategy for homology-directed SNE with different RNA-guided nucleases. On the other hand, reduced TAM recognition at weak TAM sites may require more tight base pairing between the guide sequence and the target region for successful gene editing, this may have also facilitated flexible SNE editing. This is supported by a longer guide requirement for the target flanked by the weak TAM than that by the strong TAM (Fig. 4a). In addition, we also show that the target is more sensitive to mismatches upon weak TAM. This suggests that target recognition by TnpB requires both interacting with the TAM sequence and target sequences, and the reduced recognition at the non-cognate TAM sequences can be compensated by a stronger base pairing at the target region to a certain extent.

## Methods

### Strains and growth conditions
Supplementary Data 1 contains all the information on strains, plasmids, and oligonucleotides used in this work. The genetic hosts, *S. islandicus* E233 (Δ*pyrEF*), E233S (Δ*pyrEF*Δ*lacS*)[72] and E233D1 (Δ*pyrEF*Δ*argD*) constructed in this study were derived from the original isolate, *S. islandicus* REY15A[40]. The *Sulfolobus* strains were grown at 75 °C in SVT media (basal media supplemented with 0.2% sucrose, 0.2% Tryptone, and 1% vitamin solution) or AVT media (basal media supplemented with 0.2% D-arabinose, 0.2% Tryptone, and 1% vitamin solution)[72]. 20 μg/ml agmatine or/and uracil were supplemented to the medium for the cultivation of E233, E233S, and E233D1 strains. All *E. coli* strains were grown in LB medium at 37 °C supplemented with antibiotics based on the plasmid used to be selected for. *Vibrio alginolyticus* SCSIO 43097 strain was grown in 2216E medium (Difco) at 30 °C.

### Transcriptome data analysis
The raw RNA-seq data were retrieved from NCBI GEO genomics data repository with the accession number GSE101744[73]. The FASTQ format raw data were first trimmed by Trim Galore to remove the adapter sequences and low-quality reads. Then, the trimmed reads were mapped to the *S. islandicus* REY15A genome[40] using Bowtie 2. The resulting Bam file containing reads coverage information was visualized using the Integrative Genomics Viewer (IGV)[74]. All the analyzes were performed at the Galaxy platform[75].

### Construction of mutant strains carrying TAM variants using the endogenous type IA CRISPR-Cas system
Mutant cells carrying TAM mutations at the *lacS* target site were constructed by complementing the *lacS* gene fragment containing the desired TAM mutation to the E233S strain (Δ*pyrEF*Δ*lacS*) using the endogenous type IA CRISPR gene editing system[21]. Briefly, a 37 nt protospacer sequence immediately downstream of a TCG PAM motif was selected from the mutated *lacS* gene allele of the E233S genome[72] with which two complementary oligos (lacSsp-F and lacSsp-R) were designed. Annealing the two complementary oligos yielded the spacer fragment, which was then inserted into the LguI-digested pGE1 vector[76], giving *lacS*-targeting plasmid pAC-*lacS*. The repair template contains the *lacS* gene fragment with the mutated TAM and homologous arms flanking the gene was prepared by splicing and overlapping extension (SOE)-PCR and was cloned into the pAC-*lacS* plasmid at SphI and XhoI sites, yielding pGE1-*lacS*in gene editing plasmid. 1 μg gene editing plasmid was introduced to E233S competent cells by electroporation and transformants with *lacS* insertion were selected for plasmid curing using the 5-Fluoroorotic Acid (5-FOA) counter selection. This yielded archaeal strains carrying mutated TAMs, which served as the genetic hosts for the determination of the transformation and gene editing efficiencies with the TnpB-based gene-targeting and gene-editing plasmids.

### Gene targeting assay
TnpB-based gene-targeting plasmids were constructed by cloning the DNA fragment containing the TnpB7-coding sequence, the 3′ right end (RE) and the guide sequence immediately following the RE into the expression vector, pSeSD[46]. The DNA fragment was directly amplified from the E233 genome using the primers tnpB7-F-NdeI and tnpB7-R-NheI as indicated in Supplementary Data 1. Guide sequences targeting different genomic sites were introduced to the DNA fragment using SOE-PCR. Resulting DNA fragments were inserted into the pSeSD vector at NdeI and NheI, giving gene-targeting plasmids (e.g. pSeTnpB-g*lacS*). The construction of other plasmids targeting other genomic locations followed the same procedure.

A total of 500 ng gene-targeting plasmid or non-targeting control plasmid was introduced to *S. islandicus* E233 cells or other derivative strains by electroporation[72]. A total of 20 or 50 μl electroporated cells

were plated on SVT (or AVT) plates using the two-layer plating technique. Colonies of transformants that appeared after 6 days of incubation at 75 °C were counted. The transformation efficiency was defined as colony formation units per µg plasmid DNA.

### Construction of gene-editing plasmids and determination of gene editing efficiency

Each TnpB-based gene-editing plasmid targeting the *lacS* gene was constructed by insertion of a repair template, the *tnpB7* coding sequence, the 3′ RE and the guide sequence to the pSeSD vector. The repair template carrying the mutated *lacS* gene allele (either deletion or site mutations at the target site) was prepared by the SOE-PCR using primer pairs lacSLf-SalI/lacS-SOE-Lr and lacS-SOE-Rf/lacSRr-NotI (for the *lacS* gene deletion plasmid) and inserted into the plasmid at SalI and NotI. Then the DNA fragment containing *tnpB7* coding sequence and the 3′ RE and the guide sequence was inserted to NdeI and NheI sites, giving the gene-editing plasmid pSeTnpB-g*lacS*-RT. The construction of other pSeTnpB gene-editing plasmids followed the same procedure.

Five hundred nanograms of gene-editing plasmids were introduced into *S. islandicus* E233 or each of its derivatives by electroporation, and genotypes of the colonies at the target sites were checked by agarose gel electrophoresis or sequencing of PCR products generated by colony PCR with check primers lacScheckf and lacScheckr listed in Supplementary Data 1. Noticeably, some transformants on the SVT plates showed mixed genotypes (Supplementary Fig. 16), partially due to the fact that transformants are often surrounded by satellite colonies of non-transformants that are formed due to the leaky selection of SVT media, Thus providing a challenge for the quantification of gene editing efficiency. For these reasons, 10 transformant colonies were randomly selected and mixed, serially diluted, and plated again to obtain survival cells of pure genotypes using the SVT media supplemented with uracil (non-selective media). The genotypes of colonies on the new plate were verified by either colony PCR or by X-gal staining, as indicated in the experimental workflow. Gene editing efficiency was determined by calculating the ratio of the gene-edited colony count to the corresponding total colony count.

### Gene editing assay using the dual-plasmid system

Two plasmids, pSeSD[46] and pN1dAA[54] were used in the dual-plasmid gene editing system. To generate a host for gene editing with the dual plasmid system, we deleted the *argD* gene from the E233 parental strain using the endogenous type IA CRISPR system[21]. This yielded E233D1 strain from which both *pyrEF* and *argD* genes are deleted.

In the dual plasmid gene-editing system, pSeSD was used to express a guide RNA of indicated length and provide the repair template, whereas the pN1dAA was used to express the TnpB7 protein. Specifically, the gRNA-coding sequence of different lengths were cloned into the NdeI and NheI sites of pSeSD, followed by the cloning of the repair template into the SalI and NotI sites. The TnpB7-coding sequence was cloned into the NdeI and SalI sites of the pN1dAA plasmid. These two plasmids were then co-transformed into E233D1. The transformation and subsequent quantification process followed the same procedure as described above.

### Homology-directed single nucleotide editing

Gene-editing plasmids used for introducing single nucleotide mutations are the same as those for targeted gene deletions except for the repair templates. Different from the pSeTnpB-g*lacS*-RT plasmid in which the repair template contains homologous arms flanking the *lacS* gene, the single nucleotide editing plasmid contains a repair template homologous to the target gene except for a single mismatch at the target site. The transformation and plating procedures are the same as described above. The genotypes of colonies were determined by DNA

sequencing of colony PCR products, with which the editing efficiency was calculated.

### Expression and purification of the TnpB7 RNP complex

The DNA fragment containing the coding sequence of *tnpB7* (*sire_2474*) and its 3′ flanking DNA sequences was PCR amplified from the genomic DNA of *S. islandicus* REY15A using primer pair of tnpB7-F-BamHI/tnpB7-R-NotI. The 25 nt (or 17 nt) *lacS* guide sequence was introduced to the DNA fragment using SOE-PCR with the primer pair glacS-SOEf and glacS-SOEr. The resulting DNA fragment was cloned into the expression vector, pET30a, at BamHI and NotI, yielding pET30a-*tnpB7*-*glacS* expression plasmid, which was then transformed to the *E. coli* BL21(DE3) for protein expression.

Protein expression was induced by supplementing 0.5 mM IPTG to 4 L exponentially growing culture, which was then incubated at 16 °C for 16 h with gentle shaking. Cell mass was harvested by centrifugation and cell pellets were resuspended in Buffer A (50 mM Tris−HCl, 500 mM NaCl, 25 mM imidazole, 5% glycerol, 1x protease inhibitor cocktail, pH 7.5). The resulting cell suspension was applied to a high-pressure homogenizer (JNBio), following additionally disrupted by sonication, yielding cell lysates that were centrifuged at 15,557 x *g* for 30 min for three times to remove cell debris. The supernatant was filtered through a 0.45 µm filter with a syringe and loaded to a HisTrap HP column (Cytiva). The N terminal His-tagged TnpB proteins bound to the column were washed by 25 column volumes of Buffer A (50 mM Tris-HCl, 500 mM NaCl, 25 mM imidazole, 5% glycerol, pH 7.5) and 25 column volumes of Buffer A containing 65 mM imidazole. The bound proteins were eluted by 15-column volumes of Buffer B (50 mM Tris-HCl, 500 mM NaCl, 500 mM imidazole, 5% glycerol, pH 7.5). Further purification of TnpB7 proteins was conducted by size exclusion chromatography with a Superdex 200 increase 10/300 GL column (Cytiva) running with Buffer C (50 mM Tris-HCl, 500 mM NaCl, 5% glycerol, pH 7.5). TnpB proteins eluted at 13.37 mL were combined and used for the in vitro DNA cleavage assay.

### Preparation of dsDNA substrates carrying different TAM variants

To prepare the dsDNA substrates for the DNA cleavage assay with the TnpB7-*glacS* RNP. The 25 nt target sequence from the *lacS* gene was cloned to pUC19 following a PCR protocol that amplifies the entire plasmid template with a pair of complementary primers (containing the target sequence and the 5′ TAM sequence). The PCR product was digested by the DpnI enzyme and then subjected to Gibson Assembly. This yielded the pUC19-*lacS* target plasmid, which was used as the template to introduce site mutations at the TAM sequence. We first introduced saturation mutations to each nucleotide of the TTTAA TAM. Since TnpB showed no apparent preference on all other three mutated nucleotides at −1, −4, and −5 position, only one of the representative nucleotides was selected for in the subsequent combinational mutation design. The target plasmid containing the TAM sequence (TTTAA), its derivatives carrying TAM variants or the empty vector was used as the template for PCR amplifications using a primer pair of pUC19-F1/R1, giving dsDNA substrates (1120 bp) for in vitro DNA cleavage assays.

### DNA cleavage assay

A total of 10 µl DNA cleavage reaction consists of 20 nM dsDNA substrates, 200 nM TnpB, 50 mM Tris-HCl (pH 7.5), 10 mM $MgCl_2$ and the reaction was initiated by incubation at 60 °C, 70 °C or 75 °C for indicated periods. Then the reaction was quenched by adding 2 µl 6x SDS DNA loading dye (1% SDS, 100 mM EDTA, 60% glycerol, and 0.03% bromophenol blue) and reaction products were analyzed by 1% agarose gel electrophoresis.

To identify the cleavage site of TnpB7 on the dsDNA target flanked by the TTTAA TAM, we incubated 200 nM TnpB with 2 µg linear dsDNA

target in the reaction buffer as described above. The reaction was incubated at 70 °C for 1 h. Reaction products were resolved by TBE-agarose gel electrophoresis. Two cleavage products were recovered from the agarose gel using the gel extraction kit (Omega) and sent for run-off DNA sequencing. The cleavage pattern of TnpB was determined by aligning the sequencing results with the target sequence.

The reactions used for assessing trans-cleavage activities of TnpB consist of 50 nM 5′ FAM-labeled ssDNA substrates (35 nt) or dsDNA substrate (62 bp), 300 nM TnpB7 and 50 nM activator DNA (ssDNA or dsDNA target DNA) of TnpB7. The reactions were incubated at 75 °C for 10 min. The resulting products were resolved by 15% urea poly-acrylamide gel electrophoresis and visualized with an Amersham ImageQuant 800 biomolecular imager (Cytiva).

### The *lacS* reporter gene assay

The *Sulfolobus lacS* reporter gene assay[77] was used to evaluate the promoter strength of P*araS* and its derivatives P*araS*-m38 under the basal expression condition (sucrose) and the induced condition (D-arabinose). Briefly, 10 ml exponentially growing culture containing the LacS-expressing plasmid driven by the P*araS* or P*araS*-m38 promoter was harvested and resuspended in 10 mM Tris-HCl (pH 8.0). Cell suspensions were subjected to sonication and subsequent centrifugation, yielding cell extracts for the determination of the galactosidase activity of LacS with the o-nitrophenol-β-D-galactoside (ONPG) method. Specifically, 2.5 µg total cell extracts were incubated with 2.8 mM ONPG substrate (in 50 mM $Na_2HPO_4$-$NaH_2PO_4$ reaction buffer, pH 6.5) at 76 °C for 10 min in a 400 µl reaction. Reactions were terminated by addition of an equal volume of 1 M sodium carbonate and the resulting products were quantified by measuring the absorbance of the reactions at 420 nm using a spectrometer. β-Galactosidase activity from the basal expression of the P*araS* promoter (in sucrose) was defined as 100%, with which the relative activities of other samples were calculated.

### Gene editing assay with TnpB7 in *E. coli* BL21(DE3)pLysS

Gene-targeting plasmids were constructed for the *E. coli lacZ* gene by insertion of a DNA fragment containing the TnpB-coding sequence, the corresponding 3′ reRNA-coding sequence, and a 20 nt *lacZ* guide sequence to the BamHI and NotI sites of pET30a. Repair templates containing mutated *lacZ* gene (either single-nucleotide mutation or multiple-nucleotide mutation) were inserted into the NotI site of the corresponding gene-targeting plasmid following the Gibson Assembly protocol (RK21020/ABclonal), yielding gene-editing plasmids.

These plasmids were introduced to BL21(DE3)pLysS competent cells by chemical transformation. Colonies that appeared on LB plates (30 µg/ml kanamycin and 0.2 mM IPTG) were randomly picked up, and inoculated to LB broth containing the kanamycin and IPTG of the same concentrations and incubated for 12 h at 37 °C with shaking. Then, cultures were transferred to fresh LB broth for further incubation of 8 h to facilitate the gene-editing process. Bacterial cultures were then serially diluted and plated on LB plates containing 0.2 mM IPTG, X-gal, and kanamycin. The genotypes of white and blue colonies were verified by the EcoRI digestion or the sanger sequencing of the colony PCR products. Given all the tested white colonies are proven to be gene-edited and blue colonies were shown to be unedited at the target site, the results of X-gal staining were used to calculate the gene editing efficiency of each plasmid (defined as the ratio of the white colony count to total colony count).

### Gene editing assay with TnpB7 in *Vibrio alginolyticus* SCSIO 43097

Gene-editing plasmids were constructed for the *V. algninolyticus* strain using the pBAD33-oriT vector[78] following the procedure described for the construction of *E. coli* gene-editing plasmids. The target site was selected after an ATTAA TAM of the *dgc137* gene. The RT contains

homologous flanking sequences of the target gene but carries a 900 bp deletion at the target gene. The gene editing plasmid and non-targeting control plasmid were first introduced to the *E. coli* WM3064 strain, then mobilized from the *E. coli* strain to *V. alginolyticus* strain by intergeneric conjugation[79]. Transconjugants carrying the gene-editing plasmid were inoculated to liquid 2216E medium with 30 µg/ml chloramphenicol and 0.6% L-arabinose at 30 °C for 20 h with shaking. The resulting cultures were transferred to fresh 2216E media for another 20-h incubation before being streaked onto 2216E media solidified with 1% agar. Gene editing outcomes were verified by genotyping of colonies on streak plates.

### Statistics and reproducibility

When subjected to genotyping, colonies were randomly picked up to help ensure they represent the colony population. The quantitative data in each Figures were obtained from three biologically independent experiments. No data were excluded from the analyzes. The difference between the means of different groups was analyzed with one-way ANOVA followed by the Tukey test, assuming the data is normally distributed. Data are considered as statistically significant different when the *p*-value is less than 0.05. ns means not significant ($p > 0.05$). Statistics analyzes were performed with GraphPad Prism (version 9.0).

### Reporting summary

Further information on research design is available in the Nature Portfolio Reporting Summary linked to this article.

## Data availability

The data that support the findings of this study can be found either in the article file, Supplementary Information file, or Supplementary Data file. Source data are provided with this paper. Previously published RNA-seq data employed for reRNA analysis in this study can be retrieved from NCBI GEO genomics data repository under the accession number GSE101744. Source data are provided with this paper.

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

## Acknowledgements

We thank Dr. Ling Deng at the University of Copenhagen and all members of the CRISPR and Archaea Biology Research Centre at Shandong University for stimulating discussions. This work was supported by the National Key R & D Program of China (grant number 2020YFA0906800 to QS), the National Natural Science Foundation of China (32270040 to QS and 32001022 to XF), Department of Science and Technology of Shandong Province (project number WSR2023017 to QS), Department of Education of Shandong Province (project number 2023KJ009 to XF) and SKLMT Frontiers and Challenges Project (SKLMTFCP-2023-05). Funding for open access charge: National Key R & D Program of China.

## Author contributions

X.F. and Q.S. conceived the study. X.F., R.X., XX.W., W.H., and Q.S. designed the experiments. X.F., R.X., J.L, J.Z., X.X., XN.W., J.Y., P.W., and P.Z. performed in vivo experiments. X.F. and B.Z. purified the TnpB RNP and carried out the biochemical assays. X.F., R.X., XX.W, W.H., and Q.S wrote the manuscript. All authors read and revised the manuscript.

## Competing interests

X.F., Q.S., R.X., J.L. and P.Z. are co-inventors on a patent application (CN202310186099.8) filled by Shandong University relating to this work. All other authors declare no competing interests.
