## [Peer Review File · Nature Communications]

Reviewers' Comments:

Reviewer #1:

Remarks to the Author:

General Comments:

Feng et al. report that IS605-type TnpBs in a thermophilic archaeal, *Sulfolobus islandicus* REY15A, are RNA-guided programmable endonucleases. They systematically characterized the transposon-adjacent motif (TAM) variants, and found that TnpB7 recognizes a broad range of TAMs. Then they compared TnpB7 activities on the dsDNA targets flanked by different TAM sequences in vitro and in vivo, and revealed that some TAM sequences have strong cleavage activities in vitro but induce massive cell death in vivo. In contrast, some TAM sequences show weak cleavage activities in vitro and do not interfere with cell growth in vivo. Weak TAMs facilitate efficient homology-directed targeted gene deletion. Furthermore, using weak TAMs, they successfully leveraged TnpB7 for efficient single-nucleotide editing. These findings are novel and interesting. Using weak TAMs for gene editing might be extended to other RNA-guided systems. However, the following concerns should be addressed:

Major Comments:

1. Is it possible that the minimal length of the guide-target matching region depends on different TAMs?

For example, the authors showed that a 14-15 bp gRNA-target duplex region is required for gene editing with TnpB7. However, this conclusion was based on the experiment where the weak TAM, TTTGA, was used. The authors demonstrated that successful gene editing on weak TAM by TnpB requires full sequence matching between the guide RNA and DNA target (figure 4d), and robust gene

editing was observed with mismatched gRNA on the target flanked by the strong TAM, TTTAA sequence (figure S7a). The authors should also determine the minimal length of the guide-target matching region in the presence of at least a few strong TAMs.

2. The authors studied the DNA cleavage activities of TnpB on different target flanking sequences and found that the cleavage activities of TnpB are higher than 40% on target flanked by NTTAA (figure 2d). However, a recent preprint (doi: <https://doi.org/10.1101/2023.03.06.531077>), did similar experiments and showed that the cleavage activities of TnpB are higher than 50% for TTTAA and ATTAA, but less than 5% for GTTAA and CTAA. Why are these results so different between the two studies using similar in vitro DNA cleavage assays? We advise the authors to design an experiment to differentiate between these divergent results (keeping in mind that the second manuscript is a pre-print and subject to change).

Minor Comments:

1. What are the meanings of (b) and (c and d) in Figure 4a?

Reviewer #2:

Remarks to the Author:

Reviewer #3:

Remarks to the Author:

In this work, Feng et al characterize SisTnpB7, a transposon-sourced programmable nuclease. The team focuses on understanding the sequence requirements of its targets and RNA guide that enable its activity, whether for cutting DNA or enabling gene editing in the *Sulfolobus* archaeon. As stated, there are three unknowns their work addresses: (1) Can TnpB be harnessed for homology-directed gene editing, (2) What is its specificity and (3) How does it operate in its natural host? The authors show that SisTnpB7 efficiently cuts dsDNA flanked by a range of TAMs, whose sequence variety greatly exceeds the TAM sequences predicted from the LE and RE elements of its transposon origin system. They identify a shorter reRNA that guides TnpB7 about as well as the sequence-predicted reRNA. They also show that for its native host (*Sulfolobus*), the presence of a plasmid-encoded TnpB7, reRNA and repair template increases the recovery of transformants whose sequence matches the repair template. They show that this activity is sensitive to both the target's TAM, as well as mismatches between the target and reRNA sequences.

Given that programmable nuclease activity and gene editing with TnpB enzymes is still new and filled with potential, members of the field may find this work interesting, especially TnpB7's TAM flexibility. However, its appeal is limited by the scope of this work: the gene editing potential of TnpB7 is limited (so far) to *Sulfolobus* – thus, it is unclear whether the gene editing shown in this work teaches us about TnpB7 or about specifics of DNA repair in *Sulfolobus*. We also receive a limited view of the enzyme's specificity (Fig. 4D/S7). Further, the high temperature requirements for TnpB7 really limit its utility as a genome editing tool (at least right now). Returning to the unknowns that the authors identified, they show that (1 & 3) TnpB appears to assist with homology-directed genome editing in its native host, and (2) it cuts targets with a range of TAMs. What would make this paper reach the level of quality and novelty that I associate with *Nature Communications*? First, SisTnpB7 (or one of the other TnpBs described) could be shown to work in an unrelated organism – the more divergent, the better. This might also include the templated gene editing that was described. Second, the main results should be supported with additional experiments, controls and statistics (see major concerns). Finally, the logic flow (especially in the figures) made the manuscript difficult for me to follow and would likely frustrate *NC* readers.

Major concerns:

1. Nearly all transformation/gene editing experiments were missing a key control: repair template (RT) alone (Figures 1, 3, 5). It's possible that the gene editing outcomes detected might occur even without TnpB7 activity, as seen in a wide range of microbes.
2. The use of transformation efficiency to describe TnpB activity follows transformation efficiency ~ cell death ~ cleavage by TnpB7. But is transformation efficiency a good proxy for DNA cleavage by TnpB7 in *Sulfolobus*? Show us. For example, TnpB7 with an inducible promoter could tie TnpB7 activity to cell death separately from transformation. At a minimum, show us a correlation with *in vitro* cleavage data, as seen elsewhere in the paper.
3. Gene editing efficiency, as described in this manuscript, is a challenging measure to interpret.
 - a. A part of LacS is deleted resulting in a different PCR product (finally shown in Figure S8), although it's not clear how often this outcome occurs without the RT, as GE results are never shown for 'no RT' samples.
 - b. From the methods, it sounds like colonies were not directly genotyped. Rather, 10 colonies were mixed and replated, then the genotype of those colonies checked. How does this impact the measure?
 - c. Only surviving cells are able to be genotyped. This is understandable, but should be stated nonetheless.
 - d. The method is used so many times in the manuscript, yet it likely differs from what a reader expects when seeing the term "gene editing efficiency". It should be included as a cartoon in a main figure so that readers understand the measure well.
4. As supported in the manuscript, TAM-specific differences in gene editing outcomes may be entirely due to the effect of cleavage efficiency on cell death. This could just as easily be shown by reducing TnpB7 expression levels. In which case, I don't think this finding is particularly novel. Most RNA-guided nucleases would have a similar effect if they cut less frequently, whether due to a partially-matched PAM/TAM or just lower expression levels.
 - a. Figure 3D approaches this, but we don't know how highly expressed TnpB7 is, how much this changes transformation efficiency (why isn't this shown?), how target- and TAM-specific this is, or what would happen in the opposite scenario (TTAAA TAM with low expression) . What about using another enzyme to show this isn't the case?

5. Statistics are used sporadically in the manuscript, sometimes without being defined at all (3B). I don't find cherry picking the samples you wish to compare as acceptable. For example, the selected statistic (two-tailed T-test, at least in 1E) is only used for part of the dataset without a clear reason why, and without proving (or at least stating) that data is normally distributed (a requirement for using this statistic).

a. Authors also chose N/A in regards to the statistics question in the Reporting Summary. This is not correct.

Other concerns:

1. Why didn't the authors at least compare their TAM determination method to a common method? (plasmid depletion assay - used in the original TnpB GE paper DOI: 10.1038/s41586-021-04058-1)

2. Figures are cluttered and missing annotation; this makes them difficult to interpret. Figure 1 as an example:

a. A: Individual labels are not necessary, especially given how small the writing is.

b. B: Reads of what? Why include the genome position?

c. C: Show us what we're looking at with a cartoon. Insertion site label is ambiguous.

d. D: What do the colors mean? What is SVT, and what is it selecting for? Caption is insufficient.

e. E: Different Y-axis ranges are being used within a single figure. It's not quite clear what the stats are comparing.

f. F/G: What do the colors mean? (red and orange are used at least three different ways in figure 1). While a bit helpful, the diagrams on the right are still quite confusing.

3. The importance of (0) vs (+1) guide RNAs (Fig 1) is somewhat unclear to this reviewer, especially in light of the TAM flexibility described throughout the manuscript. This result either needs more support or should be given less prominence.

4. Run-off sequencing in Fig 2B does not clearly define the cleavage site(s). The language surrounding this result should be altered and the method used for determination should be more clearly described.

5. Fig 2C would benefit from cleavage rates derived from the time-resolved cleavage data shown.

6. Fig 4D is difficult to interpret without the associated transformation efficiency data. We might just be seeing effects of low transformation efficiency (e.g. robust cleavage) as mentioned in the text. This should be shown, or at least mentioned. Same for 5B.

7. Trans-cleavage is labeled in figure S4, with no support.

8. What the point of changing the acronym from TAM to TFM? True, 'transposon adjacent motif' doesn't always make sense, but 'target adjacent motif' would be in line with the definition of PAM (protospacer adjacent motif) and require no new acronym.

9. The authors show that "the gene editing efficiency of the dual-plasmid system is lower than that of the single gene editing plasmid targeting the same target, suggesting TnpB prefers the sense-overlapping gRNA transcript." This is an interesting observation; I bet readers would be happy to see a little additional evidence and emphasis on this!

10. A summary figure would really help readers integrate these results.

---- Additional points raised by the co-reviewer ----

The manuscript 'Flexible target flanking motif requirement of TnpB enables efficient single nucleotide editing with expanded targeting scope' by Feng and co-authors addresses TnpB's DNA targeting capacity, its dependency on different TAMs, and its suitability for gene editing experiments. One of the main ideas stated in the manuscript is that TnpB exhibits different TAM requirements for cell death than gene editing. Therefore, TnpB is able to recognize more TAM sequences when used for gene editing, so it is not limited to specific genomic loci. Authors show results that claim that TTTAA is a functional TAM for TnpB7 of *Sulfolobus islandicus*; they also show that in some cases, one changed base pair in the TAM is tolerated in the gene editing process while preventing cells from dying. The authors conclude that TnpB is a potential gene editor capable of:

- Inducing DSB in DNA targets complementary to guide RNA
- Recognizing a flexible TAM (in some cases other than TTTAA) and still inducing DSB
- Using its flexible TAM recognition to make cleavage slower, and therefore give cell machinery enough time to perform homology directed repair (edit) to DNA cut by TnpB7 and allow cells to survive.

Unfortunately, the significance of this manuscript and its results are diminished for two reasons:

- It is limited to *Sulfolobus islandicus* and there is no evidence that experiments done by authors would work on any other organism – thus, the application remains unclear.
- I don't think that tolerating 'flexible' TAMs is a unique feature of TnpB – many CRISPR/Cas nucleases tolerate various PAMs.

Some minor concerns I would like to state:

- At the beginning of the manuscript, authors claim, that "TnpB proteins encoded by the IS200/605 family transposon are among the most abundant prokaryotic genes from which type II CRISPR-Cas nucleases have evolved". In my opinion, this claim is too brave since it's only a hypothesis. There are some more hypotheses regarding the evolution of different CRISPR types: theories of origins involve casposons, introns, and other transposons.
- During the manuscript, terminology changes – at first it's TAMs, then flanking sequences beyond conserved TAMs, and then – target flanking motifs (TFMs). It is quite hard to follow, I suggest stating the desired abbreviation from the beginning of the manuscript.
- In the subsection "Revealing the minimal guide RNA requirement of TnpB7-based gene editing" it is mentioned that the introduction of a potential transcriptional terminator sequence immediately downstream of the guide sequence significantly increased the gene editing activity – why is that so? I think this finding lacks explanation.

Speaking of data analysis and interpretation – the work is admirable, but some of the figures are overwhelming and hard to understand – for example, Fig. 1 a has all the TnpA/TnpB and TnpB elements written in long abbreviations, but small letters and that makes it unpleasant to read. Also, Fig. 2 c could use a simplified time scale – while trying to focus on this figure it is quite hard to hunt for the actual time scale in the figure's caption.

Speaking of methodology – my personal experience relies on different experiments, but I found this part to be following the logical flow. I think that this part was written thoroughly and clearly. Despite that, there is a statistical method mentioned in the manuscript (unpaired t-test) done for evaluating transformation efficiencies of different TnpB-based plasmids (Fig. 2 e) but no actual statistical data is shown neither in the methodology section nor in supplementary data – as a reader I find it hard to trust the significance of this experiment.

In total, I appreciate all the hard work put into the manuscript and such deep investigation of TnpB, but my already mentioned major concerns referring to the idea's applicability and uniqueness would need to be addressed before recommending this manuscript for Nature Communications.

Reviewer #4:

Remarks to the Author:

Reviewer #5:

Remarks to the Author:

In this manuscript, Feng et al. report that SiTnpB7 shows a dsDNA cleavage activity in a natural host cell. The authors claim that siTnpB7 shows flexible TAM requirements for gene editing and thus a broad range of TAM sequence. One suggested utility was that the high targeting activity when using weak TAM enables efficient single-nucleotide editing in the presence of repair template. The authors concluded that the use of different weak TAM sequences achieves gene editing with increased cell survival and can be applied for expanded targeting scopes, hoping that this result will be applicable for diverse CRISPR-Cas systems.

Overall Impression:

The authors spend too much space of this manuscript on just characterizing TAM sequence, meaning that other than the characterization of TAM, I can't find what's new in this manuscript. Plus, the authors are confused with flexibility and specificity. A flexible TAM sequence reflects a

high incidence of cleavage site on a genome, and thus a high level of off-target activity. They have attempted to test several properties of SiTnpB, but only to reach the same conclusions for DraTnpB on most of them. These are forcing me to be too reluctant to think that this manuscript deserves publication status quo.

Specifically,

1. The authors show SiTnpB7 shows more flexible TAM sequences, showing that non-canonical PAM can participate in dsDNA cleavage. However, this means that the siTnpB shows some tolerance to altered PAM sequence and thus there's possibility of high-off target activity.
2. Sequence logo analysis for PAM/TAM preference done in many CRISPR systems indicates that there are some levels of tolerance among PAM variances, This was also true for DraTnpB. In fact, the PAM sequence for SpCas9 is known to be NGG, but SpCas9 allows dsDNA cleavage somehow in the presence of NGA or NAG PAM. It's not a matter of 0 or 100, but of a degree of activity.
3. The experiments were exclusively performed in a bacterial system, thus I can't find what's useful for SiTnpB7 as bona fide genome editors. The authors mentioned that TnpB system is fascinating because of its hypercompactness. However, this statement is only true in a context of eukaryotic system, where delivery is a hot issue for gene therapy.
4. TnpB and IscB have been reported to show a short spacer sequence compared to other CRISPR systems. Nonetheless, the authors started their experiments with a long 24-nt spacer sequence, and then got back to the conclusion that the essential length of it is 14 nt. Not only the authors performed redundant unnecessary works, but failed to show a true cleavage positions in the protospacer regions. The 24-nt, and 14-nt spacer gRNA may show different cleavage profiles.
5. The authors fail to use general terminology accepted generally in the genome-editing society. Plus, they show unignorable mistakes in stating facts. For example, TnpB is known to be an ancestral gene for Type V Cas nuclease, not Type II.
6. The authors claimed that weak TAM sequence allows recovery time for the bacterial cells without compromising targeting activity. However, this claim is not convincing. Could you provide any relevant data supporting this claim? Although it is true, what's the meaning of this property in eukaryotic cells?
7. I am wondering how SiTnp7 can be used for single-nucleotide editing at a flexible position within a protospacer sequence, not confined to position 4 and 9. They failed to show this flexible single-nucleotide editing experimentally.

REVIEWER COMMENTS

Reviewer #1 (Remarks to the Author):

General Comments:

Feng et al. report that IS605-type TnpBs in a thermophilic archaeal, *Sulfolobus islandicus* REY15A, are RNA-guided programmable endonucleases. They systematically characterized the transposon-adjacent motif (TAM) variants, and found that TnpB7 recognizes a broad range of TAMs. Then they compared TnpB7 activities on the dsDNA targets flanked by different TAM sequences in vitro and in vivo, and revealed that some TAM sequences have strong cleavage activities in vitro but induce massive cell death in vivo.

In contrast, some TAM sequences show weak cleavage activities in vitro and do not interfere with cell growth in vivo. Weak TAMs facilitate efficient homology-directed targeted gene deletion. Furthermore, using weak TAMs, they successfully leveraged TnpB7 for efficient single-nucleotide editing. These findings are novel and interesting. Using weak TAMs for gene editing might be extended to other RNA-guided systems. However, the following concerns should be addressed:

Thank you for the concise summary of our work. The insightful comments and suggestions below have helped us to better present the working mechanism of TnpB in gene editing.

Major Comments:

1. Is it possible that the minimal length of the guide-target matching region depends on different TAMs?

For example, the authors showed that a 14-15 bp gRNA-target duplex region is required for gene editing with TnpB7. However, this conclusion was based on the experiment where the weak TAM, TTTGA, was used. The authors demonstrated that successful gene editing on weak TAM by TnpB requires full sequence matching between the guide RNA and DNA target (figure 4d), and robust gene editing was observed with mismatched gRNA on the target flanked by the strong TAM, TTAA sequence (figure S7a). The authors should also determine the minimal length of the guide-target matching region in the presence of at least a few strong TAMs.

We think this is an excellent suggestion. We have determined the minimal length of the guide-target matching region in the case of a strong TAM (TTAA) and found that a shorter guide length (13 nt) is required for gene editing. In addition, we found that gene editing upon another non-cognate TAM (GTTCA) also requires at least 14 bp guide-target base-pairing region, as seen on the TTTGA TAM. These new results indicate there is a cooperation between TAM recognition and target region verification during the target recognition with TnpB, i.e. upon the presence of a weak TAM, TnpB probably requires a longer guide-target duplexing region for the activation of the nuclease (a stronger base-pairing at the target region to compensate for the reduced recognition at the TAM

sequence). We have incorporated these data into the new Figure 4 and Supplementary Figure 9.

Figure 4a

Supplementary Figure 9

The revised text reads as follows on page 13: “As shown in Fig. 4c4a and Supplementary Fig. 8, upon non-cognate TAMs (TTTGA and GTTCA), a guide longer than 14 nt supported a similar activity of TnpB as the 25 nt guide sequence. The 14 nt guide sequence slightly affected the gene editing activity, and shorter guide sequences than 14 nt eliminated the activity. These results indicate the minimal requirement of guide RNA-target duplex region required for the gene editing activity of TnpB7 is 14-15 bp. Differently, we found that a 13 bp guiding sequence licenced gene editing upon the cognate TAM (TTTAA) (Fig. 4a). These results indicate the minimal requirement of guide RNA-target duplex region required for the gene editing activity with TnpB7 is 13-14 bp.”

2. The authors studied the DNA cleavage activities of TnpB on different target flanking sequences and found that the cleavage activities of TnpB are higher than 40% on target flanked by NTTAA (figure 2d). However, a recent preprint (doi: <https://doi.org/10.1101/2023.03.06.531077>), did similar experiments and showed that the cleavage activities of TnpB are higher than 50% for TTTAA and ATTAA, but less than 5% for GTTAA and CTTAA. Why are these results so different between the two studies using similar in vitro DNA cleavage assays? We advise the authors to design an experiment to differentiate between these divergent results (keeping in mind that the second manuscript is a pre-print and subject to change).

Thank you for pointing out these divergent results between the two studies. Indeed, our SisTnpB7 protein showed different PAM specificity at the “-5” position of the TAM compared to the SisTnpB1 protein as demonstrated in the preprint study. One of the noticeable differences is that the substrates used in the preprint paper are supercoiled plasmid DNAs, but linear dsDNA substrates were used in our study. To further confirm

our result, we compared the TAM specificity of TnpB7 at the -5 nucleotide with the supercoiled plasmid substrates. As shown in the attached Figure a, very similar DNA cleavage rates of plasmid targets were observed on the four plasmid substrates carrying four different TAMs.

We also compared the activity of these four TAMs in eliciting plasmid interference in which plasmids containing the native target of genome-expressed TnpB7 and different TAMs were introduced to the host. Efficient targeting by the host TnpB7 would eliminate the plasmid and lead to a reduction in colony numbers. As shown in the attached Figure b, these three TAMs of variable nucleotide in the -5 position showed a similar robust plasmid interference activity as the cognate TAM, suggesting the -5 nucleotide is relaxed for TnpB's activity.

In addition, we found that the CTTGA, ATTGA and TTTGA are equally efficient for gene editing (Figure 3), suggesting they have similar activity, reinforcing the idea that the -5 nucleotide of the TAM is a relaxed position for the activity of TnpB.

Minor Comments:

1. What are the meanings of (b) and (c and d) in Figure 4a?

The original Figure 4a is an illustration of the experimental workflow for panel (b) or panel (c) and panel (d). In the amended Figure 4, the content of panels (a), (b) and (c) were moved to the new Supplementary Fig. 8 as shown below.

Reviewer #2 (Remarks to the Author):

Reviewer #3 (Remarks to the Author):

In this work, Feng et al characterize SisTnpB7, a transposon-sourced programmable nuclease. The team focuses on understanding the sequence requirements of its targets and RNA guide that enable its activity, whether for cutting DNA or enabling gene editing in the Sulfolobus archaeon. As stated, there are three unknowns their work addresses: (1) Can TnpB be harnessed for homology-directed gene editing, (2) What is its specificity and (3) How does it operate in its natural host?

The authors show that SisTnpB7 efficiently cuts dsDNA flanked by a range of TAMs, whose sequence variety greatly exceeds the TAM sequences predicted from the LE and RE elements of its transposon origin system. They identify a shorter reRNA that guides TnpB7 about as well as the sequence-predicted reRNA. They also show that for its native host (Sulfolobus), the presence of a plasmid-encoded TnpB7, reRNA and repair template increases the recovery of transformants whose sequence matches the repair template. They show that this activity is sensitive to both the target's TAM, as well as mispairs

between the target and reRNA sequences.

Given that programmable nuclease activity and gene editing with TnpB enzymes is still new and filled with potential, members of the field may find this work interesting, especially TnpB7's TAM flexibility. However, its appeal is limited by the scope of this work: the gene editing potential of TnpB7 is limited (so far) to *Sulfolobus* – thus, it is unclear whether the gene editing shown in this work teaches us about TnpB7 or about specifics of DNA repair in *Sulfolobus*. We also receive a limited view of the enzyme's specificity (Fig. 4D/S7). Further, the high temperature requirements for TnpB7 really limit its utility as a genome editing tool (at least right now). Returning to the unknowns that the authors identified, they show that (1 & 3) TnpB appears to assist with homology-directed genome editing in its native host, and (2) it cuts targets with a range of TAMs.

What would make this paper reach the level of quality and novelty that I associate with Nature Communications? First, SisTnpB7 (or one of the other TnpBs described) could be shown to work in an unrelated organism – the more divergent, the better. This might also include the templated gene editing that was described. Second, the main results should be supported with additional experiments, controls and statistics (see major concerns). Finally, the logic flow (especially in the figures) made the manuscript difficult for me to follow and would likely frustrate NC readers.

Thank you for the comprehensive and concise summary of our study. We appreciate the insightful comments and valuable suggestions and have been able to incorporate changes accordingly throughout the manuscript. Importantly, we have successfully leveraged our archaeon-sourced TnpB for flexible gene editing in *Escherichia coli* and *Vibrio alginolyticus* (an important pathogen infecting marine animals) that grow at 37 °C and 30 °C respectively. In addition, we provided additional results showing that upon weak TAM, TnpB nuclease showed enhanced sensitivity to mismatches in the target region, compared to that upon the strong TAM, suggesting reduced recognition at the weak TAM sequence renders the target region more sensitive to mismatches. At last, we have added controls and statistics and substantially revised the Figures and main text as suggested.

Major concerns:

1. Nearly all transformation/gene editing experiments were missing a key control: repair template (RT) alone (Figures 1, 3, 5). It's possible that the gene editing outcomes detected might occur even without TnpB7 activity, as seen in a wide range of microbes.

Thank you for pointing this out. Indeed, spontaneous recombination between the repair template and the chromosome occurs sometimes, albeit at a limited frequency in *Sulfolobus* species. We have included new results in Supplementary figure 3 showing that the gene editing outcome is dependent on the catalytic activity of TnpB7, the TAM sequence and the repair template. Importantly, the dropout of any of them abolished the gene editing activity, suggesting the templated repair of DNA breaks induced by TnpB cleavage yielded gene-edited cells. We have incorporated the following text into the main text on page 6.

“Importantly, we found that the gene editing outcome is totally dependent on the catalytic activity of TnpB7, the TAM sequence and the repair template, and the dropout/mutation of any of these three factors abolished the gene editing activity (Supplementary Fig. 3), suggesting TnpB facilitates TAM- and DNA cleavage-dependent targeted gene deletion via templated DNA repair in the natural host.”

2. The use of transformation efficiency to describe TnpB activity follows transformation efficiency ~ cell death ~ cleavage by TnpB7. But is transformation efficiency a good proxy for DNA cleavage by TnpB7 in *Sulfolobus*? Show us. For example, TnpB7 with an inducible promoter could tie TnpB7 activity to cell death separately from transformation. At a minimum, show us a correlation with in vitro cleavage data, as seen elsewhere in the paper.

Thank you for the good suggestions. We agree with the point that the DNA cleavage by TnpB does not necessarily lead to a reduction in transformation efficiency, especially in the case of weak TAMs, as revealed by our study. However, upon the strong TAM (TTTAA), which is associated with a higher DNA cleavage activity in vitro, the DNA cleavage by TnpB does lead to more cell death and consistently low transformation efficiencies. Similarly, by enhancing the protein expression, TnpB also elicited robust DNA targeting and massive cell death on the target flanked by a weak TAM. These results have been incorporated into the new Fig. 5d.

3. Gene editing efficiency, as described in this manuscript, is a challenging measure to interpret.

a. A part of LacS is deleted resulting in a different PCR product (finally shown in Figure S8), although it's not clear how often this outcome occurs without the RT, as GE results are never shown for 'no RT' samples.

Thank you for pointing this out. We have incorporated new results in Supplementary Fig. 3 showing the presence of the repair template (RT) is essential for the gene editing outcome. Furthermore, sanger sequencing results of the PCR products revealed that the faster-migrating PCR product carries the gene deletion allele as the repair template.

b. From the methods, it sounds like colonies were not directly genotyped. Rather, 10 colonies were mixed and replated, then the genotype of those colonies checked. How does this impact the measure?

As explained in the method part and shown in Supplementary figure 17, certain colonies are of a mixed genotype in which the WT PCR products might derived from dead cells or competent cells. Therefore, we randomly selected 10 colonies and plated them again to quantify the ratio of surviving cells. A second-round separation (serial dilution and plating) helps the isolation of viable colonies of pure genotype. The random sampling from three independent experiments helps ensure the samples represent the colony populations.

c. Only surviving cells are able to be genotyped. This is understandable, but should be stated nonetheless.

Thank you for the suggestion. We have incorporated your suggestion in the method part. The revised text reads as follows "To quantify the ratio of edited cells, we randomly selected 10 colonies and mixed, then replated on the new plate to separate survival cells".

d. The method is used so many times in the manuscript, yet it likely differs from what a reader expects when seeing the term “gene editing efficiency”. It should be included as a cartoon in a main figure so that readers understand the measure well.

Thank you for the suggestion, a cartoon is provided in the amended Figure 1c.

4. As supported in the manuscript, TAM-specific differences in gene editing outcomes may be entirely due to the effect of cleavage efficiency on cell death. This could just as easily be shown by reducing TnpB7 expression levels. In which case, I don't think this finding is particularly novel. Most RNA-guided nucleases would have a similar effect if they cut less frequently, whether due to a partially-matched PAM/TAM or just lower expression levels.

a. Figure 3D approaches this, but we don't know how highly expressed TnpB7 is, how much this changes transformation efficiency (why isn't this shown?), how target- and TAM-specific this is, or what would happen in the opposite scenario (TTAAA TAM with low expression). What about using another enzyme to show this isn't the case?

Thank you for raising an important point here. We agree that the expression strength of the nuclease will affect the DNA cleavage, the extent of cell death and thus the gene editing outcomes. To figure out whether the gene editing outcomes upon weak TAM is TAM specific or is merely due to the reduction in the DNA cleavage activity. We have performed two sets of experiments. (1) we analyzed whether the TnpB with a reduced expression can facilitate single-nucleotide editing on the strong TAM (TTTAA). As shown in the attached Fig. 5c, a weaker expression of TnpB (about 8-fold less) still facilitates efficient gene deletion, but does not compromise transformation efficiency much when performing Single-Nucleotide editing. Importantly, a considerable fraction of cells was proven to be edited under this condition. This suggests reducing the expression strength indeed facilitated SN editing on the strong TAM. (2) We performed an experiment in the opposite scenario by analyzing the gene editing outcomes of TnpB on the weak TAM (TTTGA) upon an enhanced protein expression. As shown in attached Figure 5d, increasing the expression of TnpB (by about 6-fold) elicited an apparent DNA interference on the weak TAM site to a similar (or stronger) level as that seen on the strong TAM (under the non-induced level). Interestingly, the introduction of a repair template containing the single mismatch increased the colony numbers from 0 to more than 3700 in the case of the weak TAM, suggesting the mismatched repair template rescued the cells from consistent targeting by providing a repair template devoid of targeting. In contrast, The introduction of the same mismatched repair template only increased colony from 3.3 to 40 for the target with the strong TAM, suggestive of consistent targeting on the mismatched target. These results indicate that upon the strong TAM, TnpB shows more tolerance to mismatch at the guide-target matching region than that upon the weak TAM.

Consistent with the above results, we found TnpB requires a minimal guide-target duplexing region of 13 bp for gene editing upon the strong TAM, while in contrast, the length requirement is 14 bp for the weak TAM. The reason could be that reduced recognition at the weak TAM demands stronger base pairing at the target region, therefore less tolerance to mismatches.

We have incorporated these new results into the main text. The revised text can be found on pages 14 and 15.

5. Statistics are used sporadically in the manuscript, sometimes without being defined at all (3B). I don't find cherry picking the samples you wish to compare as acceptable. For example, the selected statistic (two-tailed T-test, at least in 1E) is only used for part of the dataset without a clear reason why, and without proving (or at least stating) that data is normally distributed (a requirement for using this statistic).

Thank you for pointing the oversight out, we have added statistics analysis results for all groups of this dataset and explained the Statistics in the figure legend and the method part. As the sample size is small (n=3), it is hard to examine whether the data belong to a normal distribution or not. Therefore, we assumed these data is normally distributed as these data were obtained from three independent experiments. We have explained this in the Statistics section in the method part.

a. Authors also chose N/A in regards to the statistics question in the Reporting Summary. This is not correct.

Thanks for pointing this oversight out, we have corrected this point in the Reporting summary file and added the "statistics and reproducibility" section in the method part.

Other concerns:

1. Why didn't the authors at least compare their TAM determination method to a common method? (plasmid depletion assay - used in the original TnpB GE paper DOI: 10.1038/s41586-021-04058-1)

Thank you for the suggestion. We have compared our method and the plasmid depletion method in the discussion part.

The revised text reads as follows on page 17: "Since most TAM/PAM sequences supporting a reduced DNA cleavage activity that is outcompeted by the host DNA repair capacity do not induce apparent DNA interference, these sequences can be overlooked by the in vivo plasmid depletion assay, a frequently used strategy to identify the PAM sequences for diverse Cas nucleases, based on their ability in depleting target plasmids upon strong PAMs. Therefore, our results hint at the possibility that the PAM/TAM sequences of different RNA-guided systems required for in vitro DNA cleavage, in vivo plasmid clearance (cell death) and gene editing are different, partly overlapped (Fig. 7), and vary between different host in which the expression level of the effector complex and DNA repair capacity are host-specific. All these factors should be considered for on-target and off-target gene editing predictions."

2. Figures are cluttered and missing annotation; this makes them difficult to interpret. Figure 1 as an example:

a. A: Individual labels are not necessary, especially given how small the writing is.

As the content of the original Fig. 1a is overlapped with that of Supplementary Fig. 1, therefore it is removed in the re-organized Figure 1.

b. B: Reads of what? Why include the genome position?

It means RNA-seq reads count mapped to the *tnpB* locus and its 3' downstream region in *S. islandicus* Rey15A host. Genome coordinates were changed to distances relative to the stop codon of the *tnpB* gene in the amended Fig. 1.

c. C: Show us what we're looking at with a cartoon. Insertion site label is ambiguous.

This panel of the figure has been updated. We have added a schematic to indicate the exact insertion site of guide sequences.

	reRNA scaffold	Guide
tnpB7-g(0)	--UUCACGCCAA----	
tnpB7-g(1)	--UUCACUGCCAA----	

d. D: What do the colors mean? What is SVT, and what is it selecting for? Caption is insufficient.

Figure 1d is amended accordingly. SVT is the media selected for the plasmid-containing cells that grow in the absence of the uracil supplement. The caption has been updated.

e. E: Different Y-axis ranges are being used within a single figure. It's not quite clear what the stats are comparing.

The ranges of the Y-axis were set to the same value. The statistics were used to compare whether the gene-editing plasmid or genome targeting plasmid has the same transformation efficiency as the non-targeting control. It was stated in the figure legend now.

f. F/G: What do the colors mean? (red and orange are used at least three different ways in figure 1). While a bit helpful, the diagrams on the right are still quite confusing.

Thank you for the feedback, we have re-organized the diagram and the figure.

3. The importance of (0) vs (+1) guide RNAs (Fig 1) is somewhat unclear to this reviewer, especially in light of the TAM flexibility described throughout the manuscript. This result either needs more support or should be given less prominence.

Thank you for the suggestion. We agree with this point that this result should be given less emphasis. Therefore, we re-organized the original Fig. 1f and removed the content of Fig. 1g and related text.

4. Run-off sequencing in Fig 2B does not clearly define the cleavage site(s). The language surrounding this result should be altered and the method used for determination should be more clearly described.

Thank you for pointing this out. The cleavage sites were labeled more clearly in the amended Fig. 2b. The revised text reads as follows on page 7 “Run-off sequencing of the cleaved products revealed TnpB7 makes a staggered cleavage centered around 15-20 bp from the TAM sequence, yielding 5' overhang (Fig. 2b and Supplementary Fig. 5)”

The method part used for the description of this experiment is also amended.

5. Fig 2C would benefit from cleavage rates derived from the time-resolved cleavage data shown.

Thanks for the suggestion, we agree that cleavage rates calculated from the time-resolved data would be more clearly showing the rate of cleavage of the TnpB enzyme upon different TAMs. However, we have found that apart from the dsDNA target cleavage, TnpB also shows trans-cleavage activities on dsDNA substrates. This means that the dsDNA substrates would be degraded into small species at longer reaction periods, making the quantification of cleavage products difficult. Therefore, we chose to show one of the representative DNA cleavage images.

6. Fig 4D is difficult to interpret without the associated transformation efficiency data. We might just be seeing effects of low transformation efficiency (e.g. robust cleavage) as mentioned in the text. This should be shown, or at least mentioned. Same for 5B.

Thank you for the suggestion. We have incorporated the transformation efficiency data related to Fig. 4D and Fig. 5B into the new Supplementary Fig. 10 and Supplementary Fig. S12.

7. Trans-cleavage is labeled in figure S4, with no support.

The result showing the trans-cleavage activity of TnpB has been included in the amended Supplementary Fig4.

8. What the point of changing the acronym from TAM to TFM? True, ‘transposon adjacent motif’ doesn’t always make sense, but ‘target adjacent motif’ would be in line with the definition of PAM (protospacer adjacent motif) and require no new acronym.

We agree with this point and have amended the TFM to TAM throughout the manuscript.

9. The authors show that “the gene editing efficiency of the dual-plasmid system is lower than that of the single gene editing plasmid targeting the same target, suggesting TnpB prefers the sense-overlapping gRNA transcript.” This is an interesting observation; I bet readers would be happy to see a little additional evidence and emphasis on this!

Thank you for the suggestion, we have performed additional experiments to investigate this difference between the single editing plasmid and the dual-plasmid system. As shown in the new Supplementary Fig. 8, the decrease in editing efficiency for the dual-plasmid system is diminished once the *tnpB7* and reRNA are expressed in the same plasmid (not as a sense-overlapping transcript). The elucidation of the underlying mechanism requires further investigations. Related text has been amended accordingly.

10. A summary figure would really help readers integrate these results.

Good suggestion! A figure illustrating the main finding of this study has been included in a new Figure (Fig. 7).

---- Additional points raised by the co-reviewer ----

The manuscript 'Flexible target flanking motif requirement of TnpB enables efficient single nucleotide editing with expanded targeting scope' by Feng and co-authors addresses TnpB's DNA targeting capacity, its dependency on different TAMs, and its suitability for gene editing experiments. One of the main ideas stated in the manuscript is that TnpB exhibits different TAM requirements for cell death than gene editing. Therefore, TnpB is able to recognize more TAM sequences when used for gene editing, so it is not limited to specific genomic loci. Authors show results that claim that TTTAA is a functional TAM for TnpB7 of *Sulfolobus islandicus*; they also show that in some cases, one changed base pair in the TAM is tolerated in the gene editing process while preventing cells from dying. The authors conclude that TnpB is a potential gene editor capable of:

- Inducing DSB in DNA targets complementary to guide RNA
- Recognizing a flexible TAM (in some cases other than TTTAA) and still inducing DSB
- Using its flexible TAM recognition to make cleavage slower, and therefore give cell machinery enough time to perform homology directed repair (edit) to DNA cut by TnpB7 and allow cells to survive.

Unfortunately, the significance of this manuscript and its results are diminished for two reasons:

- It is limited to *Sulfolobus islandicus* and there is no evidence that experiments done by authors would work on any other organism – thus, the application remains unclear.

Thank you for the comprehensive and concise summary of our work and insightful

comments.

We have tested the gene editing activity of TnpB in two unrelated bacterial hosts, *Escherichia coli* and *Vibrio alginolyticus* (an important pathogen infecting marine animals) that grow at 37 °C and 30 °C respectively. We found our archaeon-sourced TnpB also enables flexible gene editing in these bacterial organisms. This indicates the TnpB holds promise for being used in diverse microbes growing at a wide range of temperatures from 30 °C to 78 °C.

- I don't think that tolerating 'flexible' TAMs is a unique feature of TnpB – many CRISPR/Cas nucleases tolerate various PAMs.

We agree on the point that tolerating PAM/TAM variation is probably a shared feature of TnpB and certain CRISPR-Cas nucleases, as suggested by our results and previous studies. Indeed, non-cognate PAM sequences have also been found in the off-target editing sites with Cas9. This suggests non-cognate PAM sites for other Cas nucleases may also have an advantage over the cognate PAM in microbial gene editing as that seen for the TnpB system.

Some minor concerns I would like to state:

- At the beginning of the manuscript, authors claim, that “TnpB proteins encoded by the IS200/605 family transposon are among the most abundant prokaryotic genes from which type II CRISPR-Cas nucleases have evolved”. In my opinion, this claim is too brave since it's only a hypothesis. There are some more hypotheses regarding the evolution of different CRISPR types: theories of origins involve casposons, introns, and other transposons.

Thank you for pointing this typo out. We have corrected this sentence to “TnpB proteins encoded by the IS200/605 family transposon are among the most abundant prokaryotic genes from which type V CRISPR-Cas nucleases may have evolved.”

- During the manuscript, terminology changes – at first it's TAMs, then flanking sequences beyond conserved TAMs, and then – target flanking motifs (TFMs). It is quite hard to follow, I suggest stating the desired abbreviation from the beginning of the manuscript.

Thank you for the suggestion. We have amended these acronyms to TAM throughout the manuscript.

- In the subsection “Revealing the minimal guide RNA requirement of TnpB7-based gene editing” it is mentioned that the introduction of a potential transcriptional terminator sequence immediately downstream of the guide sequence significantly increased the gene editing activity – why is that so? I think this finding lacks explanation.

Thank you for raising this point. It is indeed very interesting that the introduction of a transcriptional terminator sequence after the guiding sequence affected the gene editing outcomes. One of the possible reasons could be that the 3' padding sequence of the reRNA

may have interfered with the function of the reRNA by forming intramolecular secondary structures with the guiding sequence. However, the exact underlying mechanism requires future investigations. This result has been explained in the main text.

Speaking of data analysis and interpretation – the work is admirable, but some of the figures are overwhelming and hard to understand – for example, Fig. 1 a has all the TnpA/TnpB and TnpB elements written in long abbreviations, but small letters and that makes it unpleasant to read. Also, Fig. 2 c could use a simplified time scale – while trying to focus on this figure it is quite hard to hunt for the actual time scale in the figure's caption. Speaking of methodology – my personal experience relies on different experiments, but I found this part to be following the logical flow. I think that this part was written thoroughly and clearly. Despite that, there is a statistical method mentioned in the manuscript (unpaired t-test) done for evaluating transformation efficiencies of different TnpB-based plasmids (Fig. 2 e) but no actual statistical data is shown neither in the methodology section nor in supplementary data – as a reader I find it hard to trust the significance of this experiment.

Thank you for the good suggestions. We have amended the figures and explained the statistics used as suggested. Specifically, in Fig. 1, we removed unnecessary content to the supplementary data and re-organized the figure to improve clarity. In Fig. 2c, the specific reaction time was indicated above each lane of the figure. In addition, we add a new paragraph in the method section to explain the Statistics. The statistics results have also been added to the related figures.

In total, I appreciate all the hard work put into the manuscript and such deep investigation of TnpB, but my already mentioned major concerns referring to the idea's applicability and uniqueness would need to be addressed before recommending this manuscript for Nature Communications.

Thank you again for the insightful comments and good suggestions.

Reviewer #4 (Remarks to the Author):

Reviewer #5 (Remarks to the Author):

In this manuscript, Feng et al. report that SiTnpB7 shows a dsDNA cleavage activity in a natural host cell. The authors claim that siTnpB7 shows flexible TAM requirements for gene editing and thus a broad range of TAM sequence. One suggested utility was that the high

targeting activity when using weak TAM enables efficient single-nucleotide editing in the presence of repair template. The authors concluded that the use of different weak TAM sequences achieves gene editing with increased cell survival and can be applied for expanded targeting scopes, hoping that this result will be applicable for diverse CRISPR-Cas systems.

Thank you for the concise summary of our work.

Overall Impression:

The authors spend too much space of this manuscript on just characterizing TAM sequence, meaning that other than the characterization of TAM, I can't find what's new in this manuscript. Plus, the authors are confused with flexibility and specificity. A flexible TAM sequence reflects a high incidence of cleavage site on a genome, and thus a high level of off-target activity. They have attempted to test several properties of SiTnpB, but only to reach the same conclusions for DraTnpB on most of them. These are forcing me to be too reluctant to think that this manuscript deserves publication status quo.

Specifically,

1. The authors show SiTnpB7 shows more flexible TAM sequences, showing that non-canonical PAM can participate in dsDNA cleavage. However, this means that the siTnpB shows some tolerance to altered PAM sequence and thus there's possibility of high-off target activity.

We agree with the point that the flexible TAM requirement of TnpB means there is a possibility of gene editing at other guide-matching regions flanked by non-cognate-TAM sequences, thereby the off-targeting editing predictions may have to consider this possibility. On the other hand, the expanded TAM sequences that can be used for gene editing also mean an expanded targeting scope.

2. Sequence logo analysis for PAM/TAM preference done in many CRISPR systems indicates that there are some levels of tolerance among PAM variances, This was also true for DraTnpB. In fact, the PAM sequence for SpCas9 is known to be NGG, but SpCas9 allows dsDNA cleavage somehow in the presence of NGA or NAG PAM. It's not a matter of 0 or 100, but of a degree of activity.

Thank you for the comment. We agree on the point that certain Cas nuclease also mediates DNA cleavage at non-canonical PAMs. Indeed, a weakened DNA cleavage can be expected upon non-canonical PAM/TAMs for both the TnpB and Cas9. In this study, we show that many of these weak TAMs support a similar gene on-target editing efficiency as the strong TAM but with less cell death induced. This finding on the one hand facilitates bacterial gene editing especially for homology-directed single-nucleotide editing with TnpB or other Cas nucleases. On the other hand, it contributes to the prediction of potential off-

target editing events in both prokaryotic and eukaryotic genome editing.

3. The experiments were exclusively performed in a bacterial system, thus I can't find what's useful for SiTnpB7 as bona fide genome editors. The authors mentioned that TnpB system is fascinating because of its hypercompactness. However, this statement is only true in a context of eukaryotic system, where delivery is a hot issue for gene therapy.

Thank you for pointing this out. As shown in the revised manuscript, we have leveraged our archaeon-sourced TnpB for successful gene editing in different bacterial hosts (*E. coli* and *Vibrio alginolyticus*) that grow at different temperatures. We agree with the point that the compactness of TnpB is more advantageous during the delivery process of the eukaryotic system. Therefore, we have revised the text accordingly in the main text.

4. TnpB and IscB have been reported to show a short spacer sequence compared to other CRISPR systems. Nonetheless, the authors started their experiments with a long 24-nt spacer sequence, and then got back to the conclusion that the essential length of it is 14 nt. Not only the authors performed redundant unnecessary works, but failed to show a true cleavage positions in the protospacer regions. The 24-nt, and 14-nt spacer gRNA may show different cleavage profiles.

We agree that bacterial TnpB and IscB protein functions with a 16-20 nt guiding sequence. While our SisTnpB7 protein shows ca. 30% sequence identity to DraTnpB, it might require a longer guiding sequence as the natural host grows at a much higher temperature (76 °C). This is why we start with a relatively long guiding sequence at the beginning.

To test the cleavage pattern of TnpB upon different guiding sequences of different lengths, we have performed the DNA cleavage assay using TnpB guided by the guides of different lengths. As shown in the new Supplementary Fig. 5, the cleavage position is the same for the two versions of TnpB RNPs at the target strand, but slightly differ in the NTS strand.

5. The authors fail to use general terminology accepted generally in the genome-editing society. Plus, they show unignorable mistakes in stating facts. For example, TnpB is known to be an ancestral gene for Type V Cas nuclease, not Type II.

We apologize for the typo, it is corrected in the revised manuscript. We have also changed the acronym TFM to TAM throughout the manuscript.

6. The authors claimed that weak TAM sequence allows recovery time for the bacterial cells without compromising targeting activity. However, this claim is not convincing. Could you provide any relevant data supporting this claim? Although it is true, what's the meaning of this property in eukaryotic cells?

Thank you for raising this point. As revealed by our study, weak TAM means less DNA cleavage, which would allow cells to repair the DNA lesions in time, thereby increasing the transformation efficiency. Another finding of our study is that upon weak TAM, the nuclease is more sensitive to the mismatches at the target region, this feature can be explored for the homology-directed single-nucleotide editing in both prokaryotic and eukaryotic cells. At last, our finding also facilitates the prediction of off-targeting editing by TnpB nuclease in eukaryotic cells.

7. I am wondering how SiTnp7 can be used for single-nucleotide editing at a flexible position within a protospacer sequence, not confined to position 4 and 9. They failed to show this flexible single-nucleotide editing experimentally.

Thank you for the good suggestion, we have tested TnpB's capability in mediating single-nucleotide editing across the whole target region without modifying the TAM sequence. As shown in the amended Fig. 6, TnpB facilitates a similar extent of single-nucleotide across the whole target region, confirming the flexibility of TnpB in single-nucleotide editing.

Reviewers' Comments:

Reviewer #1:

None

Reviewer #2:

Remarks to the Author:

Reviewer #3:

Remarks to the Author:

In this work, Feng et al characterize SisTnpB7, a transposon-sourced programmable nuclease. The team focuses on understanding the sequence requirements of its targets and RNA guide that enable its activity, whether for cutting DNA or enabling gene editing in the *Sulfolobus* archaeon. As stated, there are three unknowns their work addresses: (1) Can TnpB be harnessed for homology-directed gene editing, (2) What is its specificity and (3) How does it operate in its natural host? The authors show that SisTnpB7 efficiently cuts dsDNA flanked by a range of TAMs, whose sequence variety greatly exceeds the TAM sequences predicted from the LE and RE elements of its transposon origin system. They identify a shorter reRNA that guides TnpB7 about as well as the sequence-predicted reRNA. They also show that for its native host (*Sulfolobus*) and two other microbes, the presence of a plasmid-encoded TnpB7, reRNA and repair template increases the recovery of transformants whose sequence matches the repair template. They show that this activity is sensitive to the target's TAM, the expression of the system components, and mismatches between the target and reRNA sequences. They even show that a single change in the TAM enables gene editing (including DNA repair) while allowing cell survival. Given that programmable nuclease activity and gene editing with TnpB enzymes is still new and filled with potential, members of the field will likely find this work interesting, especially how TnpB7's TAM flexibility and specificity could be leveraged to direct gene editing outcomes. The appeal of this work would be even greater had the authors shown TnpB7 functioning in a mammalian model. Nonetheless, they show that the enzyme functions in non-native organisms and across a range of temperatures, which approaches this goal.

Upon comparison to the original manuscript, it is clear that the authors truly took the reviewers' comments seriously and made substantial efforts to address them. The updated manuscript is indeed much stronger and clearer. Though a flexible TAM requirement may not prove unique to TnpB7, such deep profiling and characterization of this potential genome editor illuminates its unique features, and how TnpB7 compares to already well-studied Cas nucleases. I believe that with just a few minor revisions (see below) the manuscript now reaches the level of interest and quality associated with Nature Communications.

Page Line Comment

1 31 The value for SNE could be more clear in the abstract by adding "with templated repair" or something similar to this sentence. Otherwise, the reader may be unsure if this is through a non-templated dependent (e.g. NHEJ) pathway.

3 62 Encouraging. "Demanding" is too strong of a word based on the evidence the authors provide.

3 64 It's a little ambiguous if this means all, or just some Cas12 family members. "... from which many diverse Cas12..."

F1-6 Please provide the DNA sequences for all DNA targets used in all figures in this paper (both on- and off-targets). They should be included in a supplemental table.

F1 B I think labeling 'left' and 'right' in the figure, as well as the predicted insertion site would be helpful (No need to hunt in the legend)

F2 C This panel is duplicated in the supplement. Please remove the extra instance.

7 170-9 While it's ultimately the authors' choice, I'm surprised they don't just state the TAM as TTAA rather than TTTAA. The data supports it, makes the nuclease more broadly applicable for

gene editing, and make things more obvious for readers.

7 179 This could have been concisely shown with a sequence logo.

F3 A,B,C Why represent these data in three different orientations thus making comparison across panels difficult?

ST1 This table is a nice addition to the revision 😊

F3 B ANOVA then Tukey's Test would have been a little more appropriate than an unpaired T-test.

SF7 This figure is clear and useful.

11 262 Isn't it possible that an RNA <116 nt but >95 nt could also work?

F4 These panels are nicely organized, helping the reader interpret the data.

F6 So happy to see these additions to the manuscript! The paper is much stronger with them.

17 414 Valuable insight

F7 Bottom Negligible, not "neglectable"

F7 Bottom This nice summary is a little confusing. Consider re-ordering the elements of the graphic to improve the logic flow and 'tell the story' a little better.

18 432 The methods were written thoroughly and clearly.

Reviewer #4:

Remarks to the Author:

Reviewer #5:

Remarks to the Author:

The authors have attempted to address the claims raised previously. Some of the questions were addressed properly, but the questions 3 and 6 require experimental evidence. However, the authors did not provide any experimental data, so the revised manuscript appears to show a limited improvement.

I am wondering if the TnpB system can be validated in eukaryotic system regarding the question #3. Plus, I am still curious why a slow dsDNA cleavage provides an opportunity for bacterial cells to survive. Independent of cleavage efficiency, the cells would die once dsDNA cleavage occurs. This needs to be addressed experimentally, not logically.

REVIEWER COMMENTS

Reviewer #2 (Remarks to the Author):

Reviewer #3 (Remarks to the Author):

In this work, Feng et al characterize SisTnpB7, a transposon-sourced programmable nuclease. The team focuses on understanding the sequence requirements of its targets and RNA guide that enable its activity, whether for cutting DNA or enabling gene editing in the *Sulfolobus* archaeon. As stated, there are three unknowns their work addresses: (1) Can TnpB be harnessed for homology-directed gene editing, (2) What is its specificity and (3) How does it operate in its natural host?

The authors show that SisTnpB7 efficiently cuts dsDNA flanked by a range of TAMs, whose sequence variety greatly exceeds the TAM sequences predicted from the LE and RE elements of its transposon origin system. They identify a shorter reRNA that guides TnpB7 about as well as the sequence-predicted reRNA. They also show that for its native host (*Sulfolobus*) and two other microbes, the presence of a plasmid-encoded TnpB7, reRNA and repair template increases the recovery of transformants whose sequence matches the repair template. They show that this activity is sensitive to the target's TAM, the expression of the system components, and mismatches between the target and reRNA sequences. They even show that a single change in the TAM enables gene editing (including DNA repair) while allowing cell survival.

Given that programmable nuclease activity and gene editing with TnpB enzymes is still new and filled with potential, members of the field will likely find this work interesting, especially how TnpB7's TAM flexibility and specificity could be leveraged to direct gene editing outcomes. The appeal of this work would be even greater had the authors shown TnpB7 functioning in a mammalian model. Nonetheless, they show that the enzyme functions in non-native organisms and across a range of temperatures, which approaches this goal.

Upon comparison to the original manuscript, it is clear that the authors truly took the reviewers' comments seriously and made substantial efforts to address them. The updated manuscript is indeed much stronger and clearer. Though a flexible TAM requirement may not prove unique to TnpB7, such deep profiling and characterization of this potential genome editor illuminates its unique features, and how TnpB7 compares to already well-studied Cas nucleases. I believe that with just a few minor revisions (see below) the manuscript now reaches the level of interest and quality associated with Nature Communications.

Thank you for the concise summary of our work and valuable advices on how to improve the manuscript. We have revised all the Figures, main text and supplementary file according to your suggestions.

Page Line Comment

1 31 The value for SNE could be more clear in the abstract by adding “with templated repair” or something similar to this sentence. Otherwise, the reader may be unsure if this is through a non-template dependent (e.g. NHEJ) pathway.

Thank you for the good suggestion. The abstract is amended as suggested.

3 62 Encouraging. "Demanding" is too strong of a word based on the evidence the authors provide.

Thank you for pointing this out. Amended as suggested.

3 64 It's a little ambiguous if this means all, or just some Cas12 family members. "... from which many diverse Cas12..."

Corrected as suggested.

The sentence is amended to “Two pioneer studies have reported that bacterial TnpBs encoded by the IS200/IS605 transposon represent a novel type of RNA-guided DNA endonuclease, from which many, if not all, Cas12 family members have evolved”

F1-6 Please provide the DNA sequences for all DNA targets used in all figures in this paper (both on- and off-targets). They should be included in a supplemental table.

Thank you for the good suggestion. An additional supplementary table containing all target sequences (Supplementary Data 1) has been provided along with the revised manuscript.

F1 B I think labeling 'left' and 'right' in the figure, as well as the predicted insertion site would be helpful (No need to hunt in the legend)

Good suggestion! the figure is amended as suggested.

F2 C This panel is duplicated in the supplement. Please remove the extra instance.

The duplicated content in the supplementary figure has been removed.

7 170-9 While it's ultimately the authors' choice, I'm surprised they don't just state the TAM as TTAA rather than TTAA. The data supports it, makes the nuclease more broadly applicable for gene editing, and make things more obvious for readers.

Thank you for the advice. Indeed, single nucleotide mutations at the -5 nucleotide of the TTAA TAM yield the least effect on the dsDNA cleavage activity of TnpB7. Nevertheless, the simultaneous mutation of nucleotides at the -5 and other positions of the TAM lead to slightly lower activities compared to the TAM variants with single nucleotide mutations under certain cases (for example, TTAAA (17.67%) > GTAAA (8.62%), TTTC A (45.21%) > GTTCA (27.2%), TTTC A (18.4%) > GTTAC (11.52%)), this suggests -5 nucleotide could contribute to target recognition and mutations at this nucleotide may exert an effect on the recognition of certain weak TAM variants.

Consistent with our results, structural analyses of ISDra2 TnpB complexed with substrates revealed that the TnpB interacts with all five nucleotides of a TTGAT TAM.

7 179 This could have been concisely shown with a sequence logo.

This is a very good suggestion! To generate a sequence logo, we have to define which fraction of TAM variants should be pooled for the sequence enrichment analyses. One of the most important finding of this study is that TnpB shows differing TAM requirement for eliciting DNA interference upon chromosome targeting (reduction in transformation) and for facilitating gene editing. Therefore, we sought to generate sequence logos with different activity thresholds individually. e.g. activity > 20% (robust DNA cleavage and DNA interference) or > 10% (No DNA interference, but with apparent gene editing). These newly added sequence logos were added into Supplementary table 1 and they provide a better overview of the consensus sequences that TnpB recognizes. As shown in the attached Table S1, it's clear that TnpB shows a more relaxed TAM requirement for gene editing than that for efficient DNA cleavage. In addition, our result also implicates that by increasing the TnpB expression, the TAM requirement can be further relaxed, making the TAM selection more flexible and an expanded targeting scope.

In vitro activity ^a (substrate cleaved %)	TAM sequences ^b	Sequence logo	Gene editing outcomes ^c	
27-48% (12 TAMs)	TTTAA , GTTAA , ATTAA , CTTAA , TTTCA , TATAA , GATAA , AATAA, CATAA, GTTCA , ATTCA, CTTCA	 (>27%)	Cell death for gene-targeting plasmid and gene editing plasmid	Efficient gene deletion, SNE editing may require reducing the expression level.
18.5-27% (10 TAMs)	TCTAA , ACTAA, CCTAA, GCTAA, TGTAA , AGTAA, CGTAA, GGTA, TTTTA , TTTAT	 (>18.5%)	Not tested	Not tested, Predicted to be efficient in gene editing
11.5-18.5% (10 TAMs)	TTTGA , TTTAC , TTAAA , GTTTA , ATTTA, CTTTA, GTTGA , ATTGA, CTTGA, TTTAG	 (>11.5%)	No cell death for either gene-targeting plasmid or gene editing plasmid	Efficient gene deletion and SNE.
4-11.5% (24 TAMs)	GTTAC , ATTAC, CTTAC, GTTAT, ATTAT, CTTAT, GTTAG, ATTAG, CTTAG, GTA , AATAA, CATAA, TAAA , TGA, TCAA, TATTA , TATCA , TATGA , TGTGA, TGTTA, TGTCA, TCTGA , TCTTA, TCTCA	 (>4.6%)	No cell death for either gene-targeting plasmid or gene editing plasmid	Efficient gene editing requires the overexpression of TnpB protein.

F3 A,B,C Why represent these data in three different orientations thus making comparison across panels difficult?

Thank you for the suggestion. We have re-organized the Fig. 3a and 3b, now both of them are in same orientation. The content of Fig. 3c is difficult to be aligned with that of Fig. 3a&b, therefore we keep it as its original orientation.

ST1 This table is a nice addition to the revision

Thank you for your positive feedback.

F3 B ANOVA then Tukey's Test would have been a little more appropriate than an unpaired T-test.

Thank you for the advice, we have performed the statistics analyses with one-way ANOVA and then Tukey test to compare the means of gene-targeting and gene editing groups to the Non-targeting control group. The statistics results, and corresponding text in the method part have also been updated accordingly.

SF7 This figure is clear and useful.

Thank you for your positive feedback.

11 262 Isn't it possible that an RNA <116 nt but >95 nt could also work?

This is a good question. To determine the minimal length of reRNA required for the gene editing activity of the TnpB system, we designed more truncated reRNAs with CDS-derived regions between 95 nt and 116 nt and analyzed their activities in mediating gene-deletion. As shown in updated Fig. S7b, the minimal length of the CDS-derived region of the RNA is 113 nt, meaning that the full length of the reRNA is approximately 167 nt.

F4 These panels are nicely organized, helping the reader interpret the data.

Thank you for the comment.

F6 So happy to see these additions to the manuscript! The paper is much stronger with them.

Thank you for the positive feedback.

17 414 Valuable insight

Thank you for the positive feedback.

F7 Bottom Negligible, not “neglectable”
amended as suggested.

F7 Bottom This nice summary is a little confusing. Consider re-ordering the elements of the graphic to improve the logic flow and ‘tell the story’ a little better.

Thank you for your feedback and good suggestion, we have re-organized this figure to give a better overview of the information conveyed by this study. The revised Figure 7 is shown below.

18 432 The methods were written thoroughly and clearly.
Thank you for the positive feedback.

Reviewer #4 (Remarks to the Author):

I co-reviewed this manuscript with one of the reviewers who provided the listed reports. This is part of the Nature Communications initiative to facilitate training in peer review and to provide appropriate recognition for Early Career Researchers who co-review

manuscripts.

Reviewer #5 (Remarks to the Author):

The authors have attempted to address the claims raised previously. Some of the questions were addressed properly, but the questions 3 and 6 require experimental evidence. However, the authors did not provide any experimental data, so the revised manuscript appears to show a limited improvement.

Thank you for your feedbacks and valuable suggestions on how to improve our manuscript.

I am wondering if the TnpB system can be validated in eukaryotic system regarding the question #3.

Thank you for your suggestion. While we completely agree on that it would be interesting to explore the properties of this TnpB system in mammalian cells, doing so requires some proof-of-concept experiments. As we have shown in this work, gene editing efficiencies facilitated by the TnpB are closely related to the homologous recombination repair activities of the host, which may differ from one microbe to another. More importantly, eukaryotes and bacteria employ different pathways to repair double-stranded DNA breaks (DSB). While many prokaryotes deal with the lesion exclusively with HR repair, two distinct mechanisms are known for DSB repair in eukaryotes: non-homologous end joining (NHEJ) and homologous recombination (HR), only the latter requires repair templates. In addition, while HR is preferred over NHEJ in certain yeasts, NHEJ is believed to be a default DSB repair pathway in mammalian cells. Since our study is basically about achieving better gene editing outcomes (less cell death with uncompromised editing efficiency) by maintaining the balance between the DNA cleavage activity of the TnpB nuclease and the Homologous Recombination (HR) repair activity of the host, the application of this strategy (homology-directed precision editing) in eukaryotes would require inhibit their cellular NHEJ activity. On the other hand, NHEJ-based TnpB gene editing in different eukaryotic cells was recently reported (Karvelis T, *et al.* Transposon-associated TnpB is a programmable RNA-guided DNA endonuclease. *Nature*. 2021; Xiang G, *et al.* Evolutionary mining and functional characterization of TnpB nucleases identify efficient miniature genome editors. *Nat Biotechnol.* 2023; Li Z, *et al.* Engineering a transposon-associated TnpB- ω RNA system for efficient gene editing and phenotypic correction of a tyrosinaemia mouse model. *Nat Commun.* 2024; Kutubuddin Molla, *et al.* A miniature alternative to Cas9 and Cas12: Transposon-associated TnpB mediates targeted genome editing in plants, 2024, PREPRINT). As a result, to test whether HR-based TnpB gene editing could also be applicable in eukaryotic cells would require experiments for seminar comparisons of the two approaches of gene editing, which we regard is out of the scope of the current work.

Plus, I am still curious why a slow dsDNA cleavage provides an opportunity for bacterial cells to survive. Independent of cleavage efficiency, the cells would die once dsDNA cleavage occurs. This needs to be addressed experimentally, not logically.

This is an excellent question and we apologize for not being clear with the explanation. We agree with the point that once the cleavage occurs, the cells either repair the lesions or dies. Therefore, the fate of the cells would depend on whether the nuclease activities overwhelm cells with DNA damage levels exceeding its DNA repair capacity. This conclusion has been supported by our experimental results and previous publications, which I will discuss in the following text.

As shown in the Fig. 3b, In the case of strong TAM sites that are associated with higher in vitro activities, TnpB cleavage induces massive cell death upon chromosome-targeting. With the decrease of the DNA cleavage efficiency associated with different TAMs, the transformation efficiency of gene-targeting plasmid increased gradually, until the cleavage activity is lower than that of TTTGA TAM, in which no reduction in transformation efficiency is observed anymore upon chromosome-targeting. This indicates the DNA damage levels yielded with TnpB at weak TAM sites is lower than the DNA repair capacity of the host, so that no apparent cell death occurs.

Consistently, we show in Fig. 5 that once the expression level of TnpB was enhanced, the TnpB cleavage at the weak TAM site can also lead to massive cell death, suggesting again that the fate of cells is closely related to the cellular TnpB activities.

Similar to our results, it has been reported that bacterial cells can survive weaker CRISPR-Cas9 targeting via continuous RecA-dependent homologous recombination repair activities with a sister chromosome (Cui L. and Bikard D. Consequences of Cas9 cleavage in the chromosome of *Escherichia coli*. Nucleic Acids Res. 2016) and the fate of *E. coli* cells would depend on the abundance/effectiveness of guide RNAs, meaning the balance between the DNA cleavage activity of Cas9 and homologous recombination repair activity also determines the chromosome-targeting outcomes in *E. coli* (Collias D. et al., Systematically attenuating DNA targeting enables CRISPR-driven editing in bacteria. Nat Commun. 2023).

Similar mechanism could also operate in the *Sulfolobus* species we studied. The DNA cleavage with TnpB on weak TAM sites may leave one of the chromosomal target sites intact and ready for templated repair, before the next cleavage occurs. As a result, the DNA cleavage of a reduced efficiency on the weak TAM site could be repaired by the host DNA repair machinery without inducing apparent cell death.